# The unusual structural properties and potential biological relevance of switchback DNA

Bharath Raj Madhanagopal [1], Hannah Talbot[1], Arlin Rodriguez[1], Jiss Maria Louis[1], Hana Zeghal [1], Sweta Vangaveti [1], Kaalak Reddy [1,2] & Arun Richard Chandrasekaran [1,3] ✉

Synthetic DNA motifs form the basis of nucleic acid nanotechnology. The biochemical and biophysical properties of these motifs determine their applications. Here, we present a detailed characterization of switchback DNA, a globally left-handed structure composed of two parallel DNA strands. Compared to a conventional duplex, switchback DNA shows lower thermodynamic stability and requires higher magnesium concentration for assembly but exhibits enhanced biostability against some nucleases. Strand competition and strand displacement experiments show that component sequences have an absolute preference for duplex complements instead of their switchback partners. Further, we hypothesize a potential role for switchback DNA as an alternate structure in sequences containing short tandem repeats. Together with small molecule binding experiments and cell studies, our results open new avenues for switchback DNA in biology and nanotechnology.

Molecular self-assembly using DNA allows the intricate design of nanostructures with custom shapes and sizes, programmable features, and site-specific functionalization[1–5]. DNA motifs such as the double crossover (DX)[6], triple crossover (TX)[7], paranemic crossover (PX)[8], and multi-arm DNA stars[9,10] are structural units used to construct various finite[11–13] and extended assemblies[9,14,15]. Characterization of DNA motifs and understanding their properties has contributed to the development of design principles and expansion of the DNA nanotechnology toolset to create nanometer- to micrometer-scale static structures[16,17] and dynamic devices[18,19]. To integrate a DNA motif as a structural unit in DNA nanostructures, several factors are considered: geometric parameters of the motif, thermal stability, structural robustness, polarity of the strands, and helical handedness. Chemical functionalization and design-based strategies yield nanostructures that are thermally stable[20] and nuclease-resistant[21]. Identical terminal polarities in a double helical context are typically achieved through the formation of parallel stranded DNA by specific sequence choices, DNA analogs, pH control or by the incorporation of 3′−3′ and 5′−5′ linkages that leaves two 5′ ends or 3′ ends on the DNA motifs, respectively[22–25]. To introduce left-handed helices, Z-DNA forming sequences[26] or L-DNA[27] are typically incorporated into component DNA strands. Detailed study of underlying DNA structures and new DNA motifs informs the rational design of functional characteristics in self-assembled nanostructures.

The motifs used in the field of DNA nanotechnology were initially inspired by structures that exist in nature[28]. For example, immobile 4-way junctions were inspired by Holliday junctions[29]. Double crossover molecules[6,30] have been proposed as intermediates in recombination[31], and have been demonstrated to be involved in meiosis[32]. While naturally occurring motifs can guide the design of DNA nanostructures, studying the properties of basic DNA motifs could provide insights into biological phenomena as well. There are several examples of this. PX DNA has been suggested to be involved in interactions between homologous segments of supercoiled DNA in processes such as recombination and repair[33,34]. Further, proteins that bind to PX DNA in a structure-dependent fashion have been isolated[35],

[1]The RNA Institute, University at Albany, State University of New York, Albany, NY, USA. [2]Department of Biological Sciences, University at Albany, State University of New York, Albany, NY, USA. [3]Department of Nanoscale Science and Engineering, University at Albany, State University of New York, Albany, NY, USA. ✉e-mail: arun@albany.edu

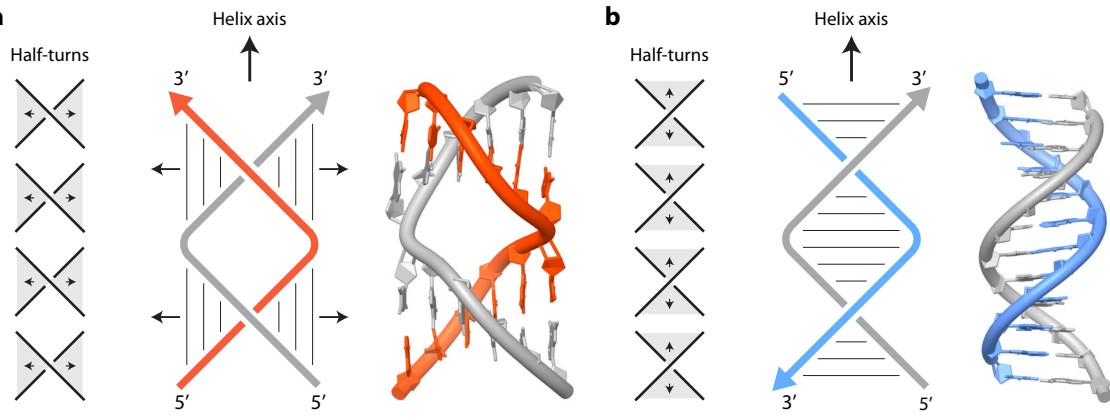

**Fig. 1 | Switchback DNA and conventional duplex.** Schematic and model of (**a**) switchback DNA and (**b**) conventional duplex. The models are adapted from PDB ID: 8EPF[41] and PDB ID: 1BNA[100], respectively. Arrows denote 3′ ends of DNA strands. The structure contains two half-turn domains, with each half-turn domain consisting of six base pairs. The switchback DNA structure has a global left-handed twist, with the helical axis of the full structure being perpendicular to the helical axis of the half-turn domains. The gray and red strands are complementary in the switchback sense.

and protein ligands that are specific to DNA motifs are also being developed[36]. In work involving the in vivo cloning of artificial DNA nanostructures, the existence of immobile Holliday junctions in *Escherichia coli* has been implied[37]. As biological materials, DNA motifs also provide useful substrates to study molecular processes such as supercoiling, crossover isomerization and structure-specific protein binding[38,39]. Thus, understanding the structural and thermodynamic characteristics of DNA motifs could help ascertain the biological implications of these structures.

In this work, we present a detailed characterization of a DNA motif called switchback DNA, first suggested by Seeman[8] and later reported as an artificial left-handed structure by Mao[40], and hypothesize its potential biological relevance. Although the motif and its self-assembly into a lattice were recently reported[41], the biochemical and biophysical properties of this molecule are unknown. The impact of the unusual left-handed topology and parallel strand orientation on the physico-chemical properties of the motif is of potential interest in DNA nanotechnology and nucleic acid structure in general. Here, we compared switchback DNA with a conventional B-form duplex using biochemical and biophysical techniques. We analyzed small molecule binding in the two structures, aided by molecular docking and studied the relative biostability of switchback DNA against five different nucleases. Moreover, we hypothesize that short tandem repeats may have the propensity to form switchback DNA as an alternate DNA structure. Our study provides a comprehensive analysis of the properties of switchback DNA, its potential role in biology and prospects in DNA nanotechnology.

## Results

### Assembly and characterization of switchback DNA

The switchback DNA motif is assembled from two DNA strands and contains a series of B-DNA half-turns aligned laterally (Fig. 1a)[40,41]. The structure shown in Fig. 1a contains two half-turns, where the helical axis of the half-turn domains is perpendicular to the axis of the full structure. Each half-turn domain consists of six base pairs, with the gray and red strands in Fig. 1a being complementary in the switchback sense. We use the term "switchback complement" for a strand that can form a switch-back structure with another strand and the term "duplex complement" for a strand that forms a conventional B-form duplex (eg: gray and blue strands in Fig. 1b). The switching back of the backbone after each half-turn results in a structure that is a globally left-handed helix with the two strands arranged parallel to each other, but the underlying half-turns are typical right-handed B-DNA with antiparallel strand orientations.

To study the properties of switchback DNA, we first used a sequence (strand A) that is known to pair with itself to form a homo-dimeric switchback structure (Fig. 2a, Supplementary Fig. 1 and Supplementary Table 1)[40]. The strands contain a thymine on the termini to prevent aggregation. To compare switchback DNA with its canonical duplex DNA counterpart (referred to as "conventional duplex" hereafter), we designed a duplex complement (strand B). We assembled the structures and validated self-assembly using non-denaturing polyacrylamide gel electrophoresis (PAGE) (Fig. 2a and Supplementary Figs. 2, 3). The formation of homodimeric switchback DNA by strand A was confirmed by the fact that it migrated similar to the conventional duplex of the same length and the absence of bands corresponding to single strands. It is to be noted that the complement to the switchback sequence can form a homodimeric switchback structure on its own due to the sequence design (Supplementary Fig. 1). We then constructed a heterodimeric switchback DNA from two unique strands (complex XY) and its corresponding duplex (complex XZ) and validated assembly using non-denaturing PAGE (Fig. 2b and Supplementary Figs. 4, 5). We observed that the switchback DNA structure was not stained on the gels as much as the conventional duplex (see later section on small molecule binding to switchback DNA and conventional duplex). To further demonstrate the formation of switchback DNA, we performed fluorescence spectroscopy experiments. We modified the 5′ end of strand X to contain a fluorescein dye and the 5′ ends of strands Y and Z to contain Iowa Black quencher (Fig. 2c). Formation of conventional duplex (complex XZ) places the fluorophore and quencher on opposite termini of the duplex, resulting in a higher fluorescence. In contrast, the fluorophore and quencher are in close proximity in the switchback DNA (complex XY) due to the parallel orientation of the strands, causing a 37% reduction in fluorescence (Fig. 2d). The sequence we used for the heterodimeric switchback DNA is derived from a DX-like motif reported recently[41], with the interacting regions containing 67% GC content. Based on our assembly of the homodimeric and heterodimeric switchback DNA as well as a recently reported crystal assembly, other sequences could also be designed for assembling a switchback DNA structure based on a modular base-pairing scheme (Supplementary Fig. 6) and containing different GC contents. Switchback DNA with 3 and 4 half-turn domains have also been created, but increasing the number of half-turns further may induce aggregation of the structures[40].

We then characterized the switchback DNA and the corresponding duplexes using circular dichroism (CD) spectroscopy and UV melting.

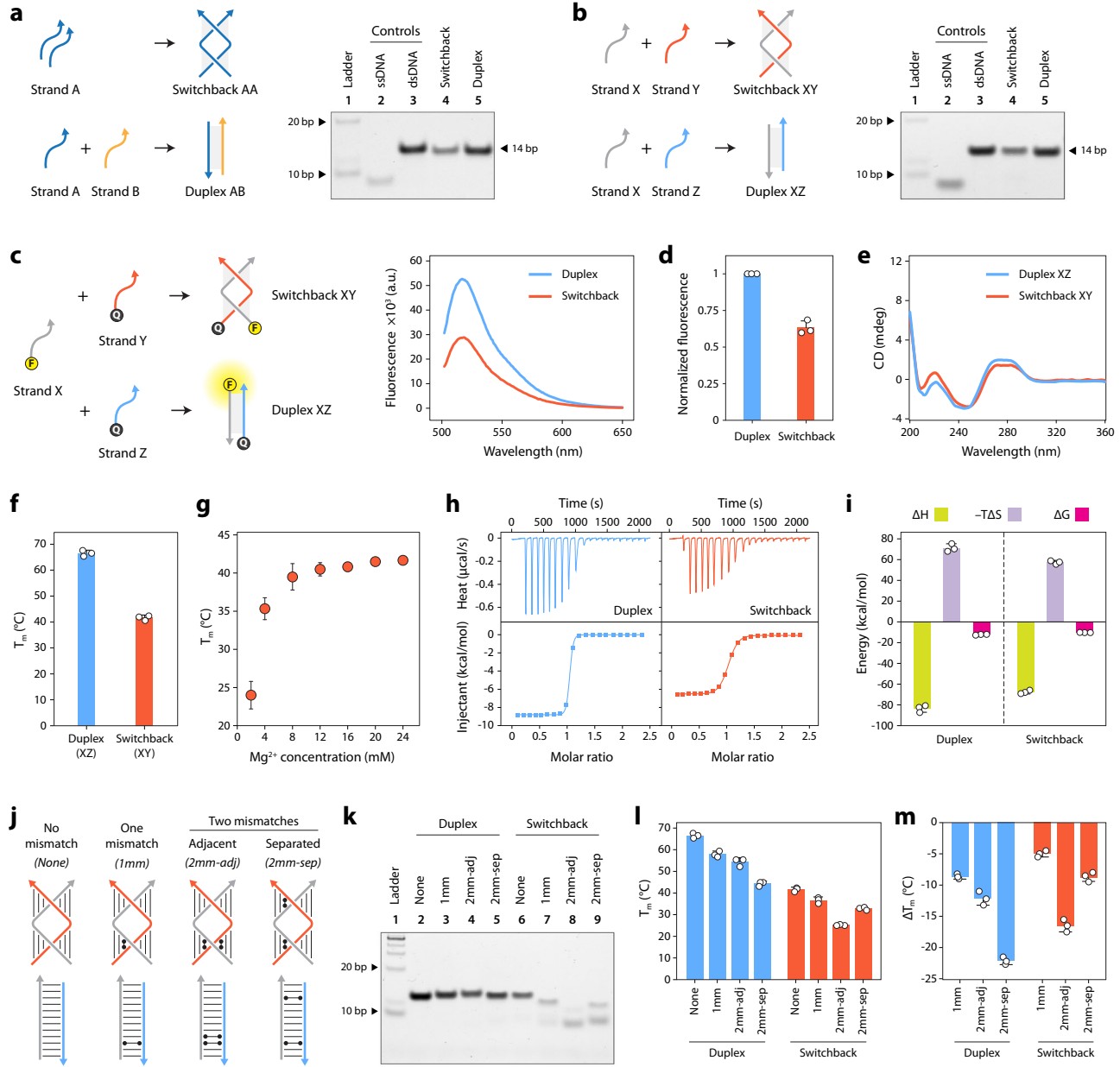

**Fig. 2 | Assembly and characterization of switchback DNA. a** Non-denaturing polyacrylamide gel electrophoresis (PAGE) analysis of homodimer switchback DNA. **b** Non-denaturing PAGE analysis of heterodimer switchback DNA. **c** Fluorescence experiments showing the formation of switchback DNA with strands in parallel orientation. **d** Percent fluorescence reduction during switchback DNA formation. **e** CD spectra of conventional duplex and heterodimeric switchback DNA. **f** UV thermal melting temperatures (Tm) of conventional duplex and switchback DNA. **g** Melting temperature of heterodimer switchback DNA at different Mg$^{2+}$ concentrations. **h** Isothermal titration calorimetry (ITC) thermograms of conventional duplex and heterodimer switchback DNA. **i** Thermodynamic parameters of conventional duplex and switchback DNA. **j** Scheme of switchback DNA and conventional duplex with 1 or 2 mismatches. **k** Non-denaturing PAGE analysis of structures with mismatches. **l, m** Melting temperatures and difference in melting temperature compared to perfectly matched sequences for conventional duplexes and switchback DNA. Data represent mean and error propagated from standard deviations of experiments performed in triplicates ($n$ = 3 independent experiments). The data shown in (**a**), (**b**) and (**k**) are representative of experiments performed multiple times ($n$ > 3) with similar reproducible results. Source data are provided as a Source Data file.

The CD signature of switchback DNA was similar to that of its conventional duplex, with a positive band at 270 nm and a negative band at 250 nm, confirming that the half-turns in the switchback DNA resemble a typical B-DNA structure despite the overall left-handed nature of the switchback structure (Fig. 2e and Supplementary Fig. 7). The CD spectra we obtained is also consistent with that reported for the homodimer switchback DNA[40]. Thermal melting analysis showed that both the homodimeric and heterodimeric switchback structures had lower melting temperatures (Tm) compared to their duplex counterparts (Fig. 2f and Supplementary Fig. 8), with the switchback DNA requiring at least 10 mM Mg$^{2+}$ to be stable (Fig. 2g). In switchback DNA, the juxtaposition of the half-turns probably necessitates the presence of divalent ions to screen the negative charges on the backbone, akin to multi-crossover structures arranged in bundles or square lattices that also require similar Mg$^{2+}$ concentrations to remain stable[42,43].

## Thermodynamics of switchback DNA formation

We studied the thermodynamics of switchback DNA and conventional duplex formation using isothermal titration calorimetry (ITC) by comparing the hybridization between the two component strands in

each structure (Fig. 2h). We measured a ΔG of −12.23 kcal/mol for the conventional duplex (with a Kd of 1.3 nM), showing that this structure was thermodynamically more stable than its switchback counterpart with a ΔG of −10.26 kcal/mol and a Kd of 30.7 nM (Fig. 2i). A comparison of the ΔH of the two structures (−83.33 kcal/mol for conventional duplex and −67.61 kcal/mol for switchback) revealed that although the number of base pairs is the same for the two structures, the enthalpy that contributes to the stabilization of the structure is relatively less for switchback DNA (Supplementary Table 2). While the formation of both conventional duplex and switchback DNA is enthalpy driven and duplex is relatively more stable, our results show that the entropic penalty for forming the duplex is higher than that for forming the switchback, which suggests that there are degrees of freedom in the switchback that don't exist in the duplex. These differences are possibly due to the disruption in the continuous base pairing and the elimination of a base stack in the middle.

## Mismatch tolerance

We then investigated the contribution of the half-turn domains to the overall stability of switchback DNA by introducing mismatches in the component DNA sequences. A previous study tested the stability of such structures using homodimer complexes which restricted the analysis to pairs of mismatches[40]. Here, we evaluated how the number (one or two mismatches) and the location of mismatches (one or both half-turn domains) affect the stability of the structure. To test this, we designed three variations of the heterodimeric switchback DNA: (1) a single mismatch in one domain (1 mm), (2) two adjacent mismatches in one domain (2 mm-adj) and (3) two mismatches spread out between the two domains of switchback DNA (one mismatch per domain) (2 mm-sep) (Fig. 2j and Supplementary Fig. 9). For each of these cases, we also designed duplex complements to form conventional duplexes with one or two mismatches. Non-denaturing PAGE analysis showed that the formation of the conventional duplex was not affected by the presence of one or two mismatches, as indicated by the appearance of the band similar to the control structure without any mismatches (Fig. 2k, lanes 2–5). For the switchback structure, the presence of one mismatch slightly affected the assembly. We observed a band corresponding to the switchback DNA with a single mismatch (lane 7) that migrated closer to the position of the structure without any mismatches (lane 6) and is different from the band corresponding to single strands, indicating that the structure is still predominantly intact (Supplementary Fig. 10). However, the formation of the switchback DNA was more affected when two mismatches were introduced (Fig. 2k, lanes 8–9). Further, assembly was most affected when the two mismatches were placed adjacent to each other, indicating that both half-turn domains need to be stable to hold the full structure intact. We then measured the impact of the mismatches on the thermal stability of the structures (Fig. 2l). As seen with the PAGE experiment, UV melting studies also showed that the introduction of one mismatch is tolerated in a switchback structure but when two mismatches were introduced the position of the mismatch determined the stability of the structure (Fig. 2m and Supplementary Table 3). In these examples, we chose the location of the mismatches to be in the middle of the half-turn domains, away from the point where the strands switch back into the next half-turn domain, and note that the specific location of the mismatch within the half-turn domains may have different effects on the stability of the structure.

## Structural preference between switchback DNA and conventional duplex

Next, we investigated the preference of a DNA sequence to form switchback DNA or conventional duplex when presented with a switchback complement or a duplex complement. We used longer duplex complements for these experiments to distinguish the conventional duplex from switchback DNA on non-denaturing gels

(Supplementary Figs. 1, 2, 4). First, we studied competition between the switchback complement and the duplex complement by annealing the homodimeric switchback sequence (strand A) and its duplex complement (strand B) in different molar ratios (Fig. 3a and Supplementary Fig. 11). In the absence of the duplex complement, a homodimer switchback DNA is formed. As the ratio of the duplex complement is increased, the band corresponding to the conventional duplex increases and attains maximum yield at a molar ratio of 1:1, showing that a switchback sequence prefers to form a conventional duplex in the presence of the duplex complement. Due to the sequence design of switchback structures, the duplex complement can form a homodimeric switchback structure of its own, as observed with higher ratios (i.e., excess) of strand B. Normalized assembly yields showed that the strands exhibit an absolute preference for duplex (complex AB) rather than switchback DNA (complexes AA or BB), matching the expected values for maximum duplex formation at different strand ratios. We then performed a similar experiment for the heterodimeric switchback sequence (strand X) in the presence of both its switchback complement (strand Y) and duplex complement (strand Z) (Fig. 3b and Supplementary Fig. 12). Again, at a ratio of 1:1:1, where 1 molar equivalent of both the switchback and duplex complements (Y and Z) were available for X, nearly all of X paired with Z to form the conventional duplex (complex XZ).

Since these observations show an absolute preference for the duplex over switchback DNA when the two competing strands are present during assembly, we then tested if the duplex complement could displace its counterpart in pre-assembled switchback DNA and convert it into a conventional duplex. We assembled homodimeric switchback DNA (complex AA) and added increasing concentrations of the duplex complement (strand B). As the concentration of the duplex complement increased, there was a concomitant increase in the amount of duplex present in the solution while the amount of switchback DNA decreased (Fig. 3c and Supplementary Fig. 13). Nearly all the complexes in the solution were conventional duplexes when the duplex complement was present in 1.25 molar equivalents. Excess of the duplex complement forms a switchback complex of its own (complex BB). We observed similar results with the heterodimer switchback DNA (complex XY), with more switchback complexes converted into conventional duplexes with an increase in the concentration of the duplex complement (strand Z) (Fig. 3d and Supplementary Fig. 14). These results indicate that a duplex complement can displace a switchback complement from a pre-assembled switchback DNA structure, a phenomenon that is essentially toehold-less DNA strand displacement based on structural stability and not just sequence affinity. Overall, our results show that component strands have a preference towards conventional duplexes even in the presence of a switchback complement, both during and post-assembly.

## Small molecule binding to switchback DNA

The study of small molecule binding to synthetic DNA motifs and nanostructures plays a major role in the design of DNA nanostructure-based drug delivery vehicles[44–48]. In that context, we analyzed the binding of two classes of small molecules (intercalators and groove binders) to switchback DNA and conventional duplex. As representative intercalators, we used ethidium bromide (EBr) and GelRed, and as representative groove binders, we used Hoechst 33258 (H33258) and 4′,6-diamidino-2-phenylindole (DAPI). These small molecules are known to exhibit enhanced fluorescence when bound to DNA[49–51]. We incubated the homodimeric switchback DNA and its corresponding conventional duplex with different concentrations of each small molecule and measured the fluorescence signal at their characteristic emission wavelengths (Supplementary Fig. 15). We observed that the fluorescence of small molecules upon binding to the conventional duplex was about 1.5–2 times higher than those bound to switchback DNA in the concentration ranges we tested (Fig. 4a, b and

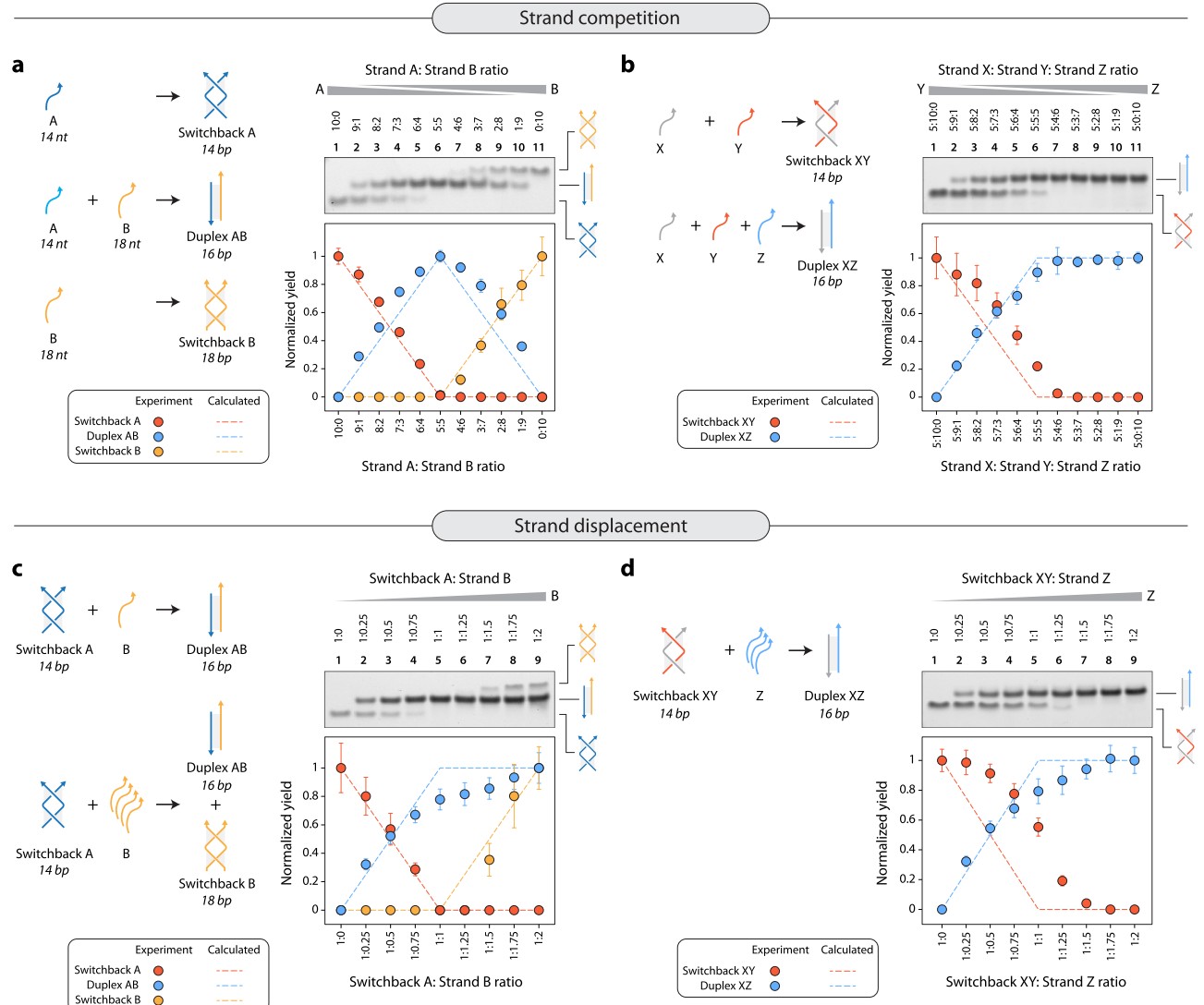

**Fig. 3 | Structural preference between conventional duplex and switchback DNA. a** Assembly of homodimer switchback formed by strand A in the presence of its duplex complement B. Non-denaturing PAGE shows the formation of the homodimer switchback in the absence of its duplex complement, and increase in duplex population as the ratio of the duplex complement in the solution increases. Excess strand B forms a homodimer switchback of its own. The quantified result of the gel bands shows a decrease in switchback DNA and an increase in duplex, reaching a maximum yield at 5:5 ratio when the strands are in equal quantities. **b** Assembly of heterodimer switchback formed by strands X and Y in the presence of its duplex complement Z. **c** Addition of duplex complement (strand B) to a pre-assembled homodimer switchback (complex A) causes displacement of one of the strands in the switchback DNA, resulting in duplex formation. **d** Addition of duplex complement (strand Z) to a pre-assembled heterodimer switchback (complex XY) causes displacement of strand Y in the switchback, resulting in duplex formation (complex XZ). Data represent mean and error propagated from standard deviations of experiments performed in triplicates. Source data are provided as a Source Data file.

Supplementary Fig. 16). The observed difference in fluorescence enhancement is probably a consequence of fewer number of base stacks and interrupted grooves in switchback DNA compared to the regular duplex.

We then performed molecular docking to obtain snapshots of how known intercalators and groove binders interact with switchback DNA compared to the conventional duplex. For the intercalator, we observed that the binding mode of EBr to the switchback DNA is similar to that of the conventional duplex. Each half-turn domain of the switchback can accommodate intercalators the same way as a duplex, without significant disruption of the structure (Fig. 4c). For the groove binder H33258, we observed that the first two binding sites with the highest docking scores occurred in each half-turn domain of the switchback DNA (Fig. 4d). Further analysis showed that H33258 molecules prefer to bind to minor grooves but may also bind to major grooves if the minor grooves are already occupied. The continuous

stretch of DNA in the duplex allows for more than one H33258 molecule in the minor and major groove, while the switchback DNA, with its two stacked half-turns, is unable to accommodate more than one H33258 molecule in either groove (Supplementary Fig. 17).

## Biostability of switchback DNA

One important consideration for using DNA nanostructures in biological applications is their ability to withstand degradation by nucleases[52]. Here, we compared the biostability of switchback DNA and conventional duplexes by testing them against a variety of nucleases (Fig. 5a). We designed our experiments to be performed at the physiological temperature of 37 °C and in buffer conditions typically optimized for best enzymatic activity (Supplementary Table 4). First, we incubated both the structures with different amounts of DNase I, a commonly used endonuclease, at 37 °C for 1 hour and analyzed the nuclease-treated samples using a gel-based method we had

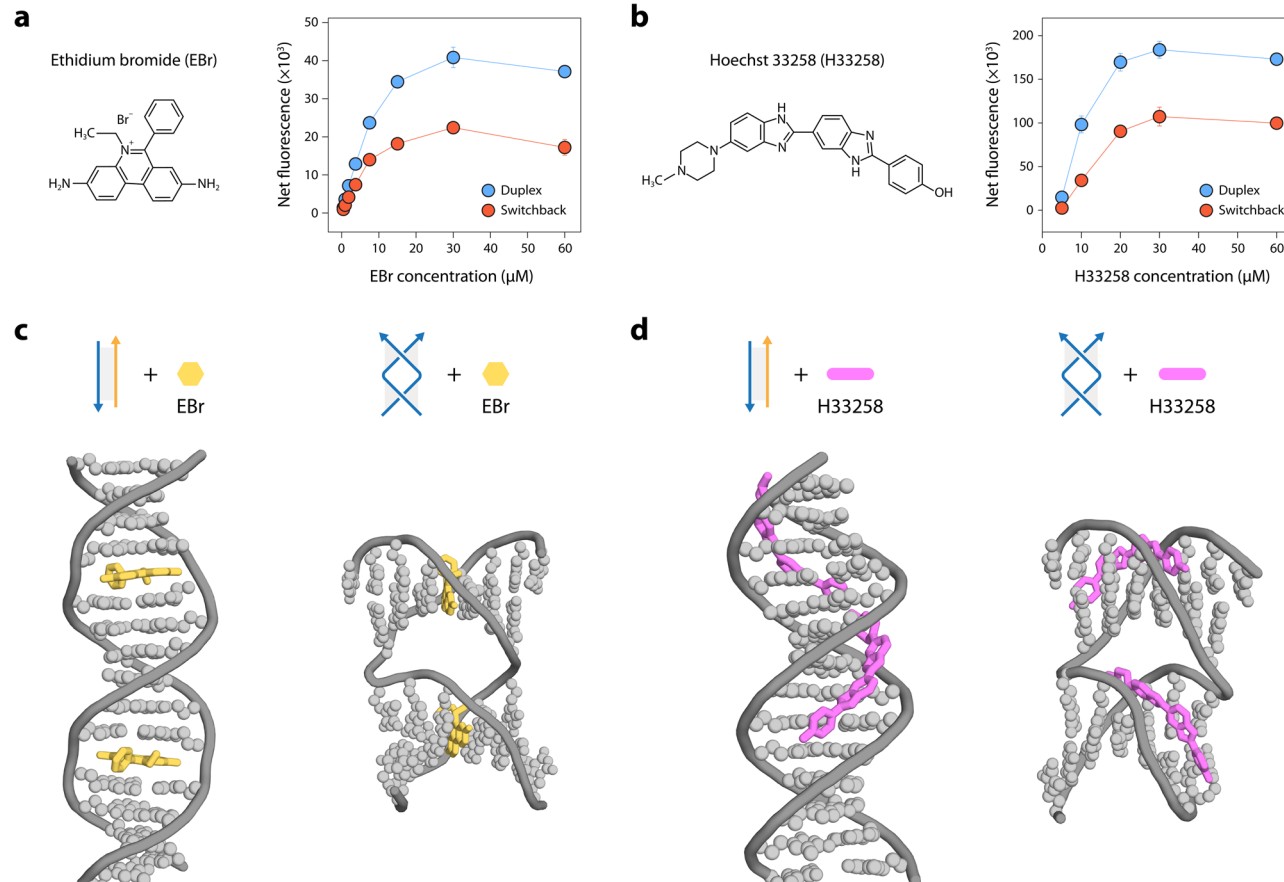

**Fig. 4 | Binding of small molecules to switchback DNA. a, b** Fluorescence intensities of conventional duplex and switchback DNA with different concentrations of ethidium bromide (EBr) and Hoechst 33258 (H33258), respectively. Data represent mean and error propagated from standard deviations of experiments performed in triplicates. **c, d** Molecular docking analysis of conventional duplex and switchback DNA with EBr and H33258, respectively. For EBr, two representative EBr-bound intercalation sites are shown. For H33258, the bound molecules for the top two binding modes are shown. Source data are provided as a Source Data file.

established before[53,54]. While both structures degraded with higher amounts of DNase I, the switchback DNA showed lower degradation compared to conventional duplexes (Fig. 5b and Supplementary Fig. 18). Kinetics of nuclease degradation showed that the switchback structure degraded more slowly compared to the conventional duplex when treated with DNase I (Fig. 5c and Supplementary Fig. 19).

We then expanded our analysis to other nucleases. We incubated the switchback DNA and conventional duplex with different concentrations of T5 exonuclease (T5 exo), exonuclease V (Exo V), exonuclease VIII (Exo VIII), and micrococcal nuclease under the optimal conditions for each enzyme (Fig. 5a, b and Supplementary Fig. 18). We then analyzed the degradation kinetics against these nucleases at one enzyme concentration (Fig. 5c and Supplementary Fig. 19). Nucleases are known to have different activities, mechanisms, substrates, and polarities of digestion when acting on double-stranded DNA. Similarly, we observed that different nucleases had varying activity levels on switchback DNA (Fig. 5d). In a 1 h assay, DNase I, micrococcal exonuclease, and Exo V almost fully digested the switchback DNA with ~1 unit enzyme while it required ~2 units of T5 exonuclease and >10 units of Exo VIII to fully digest switchback DNA. The relative biostability of switchback DNA compared to the conventional duplex also differed for these enzymes. In DNase I and T5 Exo, the switchback DNA showed higher biostability compared to duplex, whereas against Exo V, Exo VIII, and micrococcal nuclease, the duplex showed higher biostability (representative data at 8 min time point is shown in Fig. 5e). Our results show that nuclease activity on DNA motifs can be structure dependent,

with different enzymes having varying effects on the two DNA structures. These results hold promise in DNA nanotechnology, where enhanced biostability could be introduced through structural design that occludes enzyme binding at specific loci and strategic placement of crossovers[21,55,56].

## Effect of switchback DNA on cell viability and immune response

With future biological applications in mind, we tested the cell viability and immune response for switchback DNA using HeLa cells as a model cell line. As cell experiments are performed at the physiologically relevant temperature of 37 °C, we first tested the stability of switchback DNA and showed that the structure is intact for at least 48 h at 37 °C in buffer (Supplementary Fig. 20). We performed the MTT assay to test cell viability and observed only a marginal reduction in cell viability after 48 h for switchback DNA compared to the conventional duplex, showing that the switchback DNA structure is not harmful to the cells (Fig. 6a). Brightfield microscopy images of the cells confirmed that switchback DNA did not interfere with cell growth when compared to cells incubated with phosphate buffered saline (PBS), 1× TAE-Mg$^{2+}$ or untreated cells (Fig. 6b and Supplementary Fig. 21). To examine the immune response, we incubated HeLa cells with different concentrations of switchback DNA and conventional duplex for 24 h, isolated cellular RNA and performed RT-qPCR for three candidate markers of immune response (CXC motif chemokine ligand 8 (*CXCL-8*), chemokine (C-C motif) ligand 5 (*CCL5*), and interferon induced protein with tetratricopeptide repeats 1 (*IFIT1*). For comparison, we used 1×

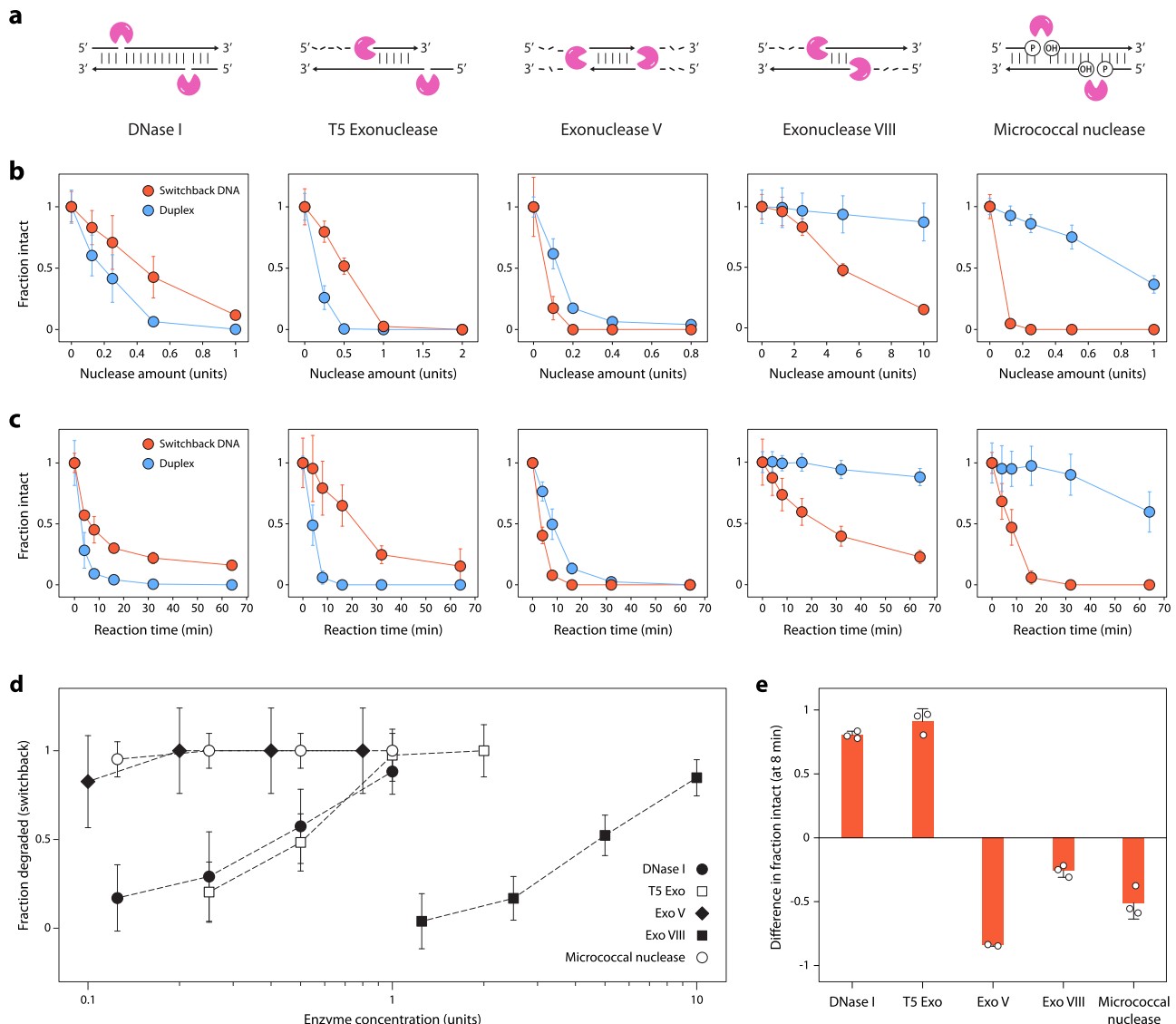

**Fig. 5 | Biostability of switchback DNA. a** Known activity on duplexes of nucleases used in this study. **b** Degradation trends of switchback DNA and conventional duplex with different concentrations of each nuclease. **c** Timed analysis of switchback DNA and conventional duplex with different nucleases (DNase I: 0.75 U, T5 Exo: 1 U, Exo V: 0.5 U, Exo VIII: 10 U, micrococcal nuclease: 0.25 U). **d** Activity of different nucleases on switchback DNA. **e** Comparison of relative intact fractions of switchback DNA and conventional duplex at the 8 min time point from data shown in (**c**). Data represent mean and error propagated from standard deviations of experiments performed with a minimum of two replicates. Source data are provided as a Source Data file.

TAE-Mg$^{2+}$ buffer as negative control and polyIC RNA, a synthetic stimulant known to induce immune response, as a positive control (Supplementary Fig. 21). We observed that the expression level of *CXCL-8* was significantly lower (3- to 7-fold lower) when treated with switchback DNA than when treated with conventional duplex across the concentration range of 100–1000 nM of DNA (Fig. 6c). We did not observe substantial differences in the levels of the other two markers between switchback DNA and conventional duplex (Fig. 6c), whereas polyIC treatment induced *CCL5* expression (Supplementary Fig. 21). Overall, our results show that switchback DNA induced lower expression of immune response markers compared to duplex DNA or polyIC in HeLa cells for the markers tested here. These findings suggest that switchback DNA could potentially be a low-immunogenic alternative to duplex-based nanostructures for biological applications.

### Potential biological relevance of switchback DNA
The modular base-pairing scheme of switchback DNA is distinct from the conventional end-to-end base pairing model. While molecular recognition between nucleic acids mainly occurs via the conventional base pairing model, it is conceivable that the switchback DNA mode of interaction may also exist in certain structural contexts between pairs of sequences that allow such pairing. We hypothesize that switchback DNA could be a potential alternate structure in short tandem repeat sequences. That is, a pair of sense and antisense strands containing tandem direct repeat sequences could form either the switchback DNA or a conventional duplex (Fig. 7a). We screened various direct repeat sequences involved in repeat expansion diseases and observed that in some cases, a given sequence could have the same switchback complement and duplex complement (Supplementary Fig. 22). Expansion of short tandem repeats (stretches of 2-12 bp long repeating tracts of DNA) found in both coding and noncoding regions of the genome causes over 50 neurological, neuromuscular, and neurodegenerative diseases[57]. Many of these repeat sequences have been shown to form non-canonical DNA structures such as hairpins [e.g., spinocerebellar ataxia type 2, (CAG)$_n$][58], G-quadruplexes [e.g., *C9orf72* amyotrophic lateral sclerosis-frontotemporal dementia, (GGGGCC)•(GGCCCC)][59],

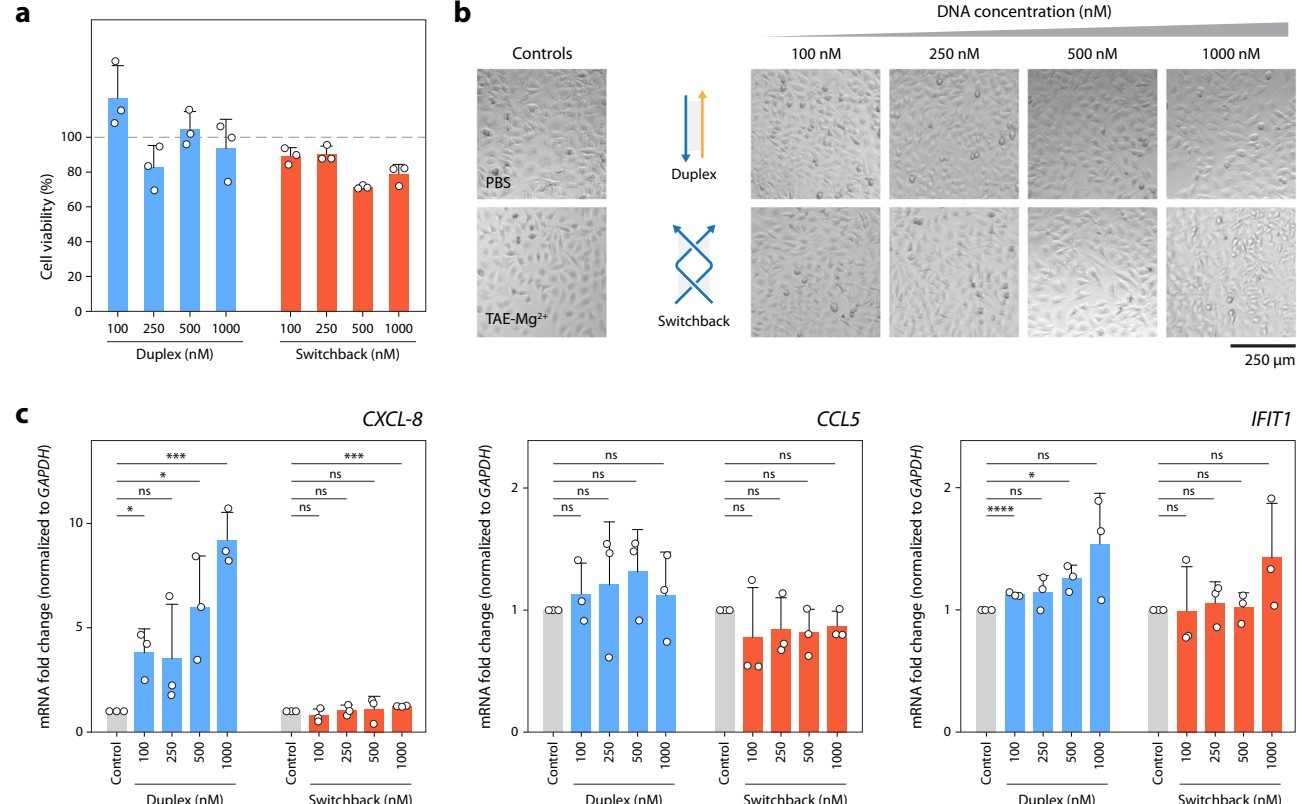

**Fig. 6 | Effect of switchback DNA on cell viability and immune response.**
**a** Viability of cells treated with conventional duplex and switchback DNA at different DNA concentrations. Data is normalized to viability of cells treated with 1× TAE-Mg$^{2+}$ (gray dashed line). **b** Brightfield microscopy images of cells treated with different concentrations of conventional duplex and switchback DNA. **c** RT-qPCR analysis of immune response markers, *CXCL-8* (Duplex, from left-to-right: $P = 0.0135$ ; $P = 0.1711$; $P = 0.0255$; P = 0.0004, Switchback, from left-to right: $P = 0.3162$; $P = 0.802$; $P = 0.7869$; $P = 0.0001$), *CCL5* (Duplex, from left-to-right:

$P = 0.4245$; $P = 0.5294$; $P = 0.1975$; $P = 0.5967$, Switchback, from left-to right: $P = 0.4005$; $P = 0.3549$; $P = 0.1692$; $P = 0.1367$), and *IFIT1* (Duplex, from left-to-right: $P < 0.0001$; $P = 0.1362$; $P = 0.0129$; $P = 0.0885$, Switchback, from left-to right: $P = 0.9643$; $P = 0.5991$; $P = 0.7835$; P = 0.1766). Data represent mean and standard deviation from three biological replicates. Unpaired two-tailed *t*-test was used to compare switchback DNA and conventional duplex treatments to controls: ns−not significant, *$P < 0.05$, ***$P < 0.001$, ****$P < 0.0001$. Source data are provided as a Source Data file.

triplexes [e.g., Friedreich's ataxia, (GAA•TTC)][60], Z-DNA [e.g., skeletal displasias, (GAC)$_n$][61], and slipped DNA junctions [e.g., myotonic dystrophy type 1, (CTG)•(CAG) and fragile X syndrome, (CGG)•(CCG)][62]. These alternate structures can influence DNA repeat instability, transcription, and protein binding[63].

To study the relevance of switchback DNA in tandem repeat sequences, we chose the trinucleotide repeat (CAG)$_4$ and its complement (CTG)$_4$ associated with Huntington's disease, various spinocerebellar ataxias, myotonic dystrophy type 1, and Fuchs endothelial corneal dystrophy[64]. We performed molecular dynamics (MD) simulations to study the conformational dynamics of the two structures formed by these same pair of strands (Fig. 7a, b). Our simulations indicate that once formed, the overall structure of the switchback DNA is well tolerated by the repeat sequences, as indicated by only a marginally higher root mean square deviation throughout the simulations compared to the duplex (Supplementary Fig. 23). Next, we calculated the fluctuations of the nucleotides in the two structures. Higher fluctuation is suggestive of weaker interactions leading to local instability. We observed that the fluctuations of the two structures are higher on the terminal nucleotides, which is expected. However, switchback DNA shows consistently higher fluctuations compared to the duplex, which suggests localized regions where the structure is more likely to unfold compared to the duplex (Fig. 7c). Additionally, we used the Molecular Mechanics/Poisson Boltzmann Surface Area (MMPBSA) method[65] to calculate thermodynamic properties of the two configurations within the repeat sequence, from the simulation trajectories. The predicted

$\Delta H$ values for the duplex and the switchback conformations differ only by ~12% (Fig. 7d), which primarily arises from altered electrostatics and solvation between the two conformations (Supplementary Table 5). This indicates that both conformations are energetically feasible even if the duplex conformation is slightly more favorable.

Next, we sought to test whether CAG/CTG repeats could adopt switchback conformation. Since the formation of antiparallel conventional duplex is preferred over parallel switchback DNA for the two complementary strands, we imposed geometric constraints on the strands by fixing them on a DNA double crossover (DX) scaffold that places the two sequences next to each other in parallel orientation (Fig. 7e). The DX motif contains two adjacent double-helical domains connected by two crossover points, a motif well characterized in our group and several others[6,66–70]. We extended one strand from each double helical domain to create single-stranded "interacting regions", where both extensions are 5′ to 3′ from the DX scaffold. The design also contains a TT linker between the DX scaffold and the interacting regions to allow some flexibility. We fixed interacting region 1 to contain (CAG)$_4$ and varied the sequences in interacting region 2 to contain (i) (CTG)$_4$ that can form a switchback with (CAG)$_4$ in interacting region 1, (ii) (CTG)$_2$T$_6$ that can form only one domain of the switchback DNA or (iii) a T$_{12}$ sequence that cannot recognize (CAG)$_4$ (Fig. 7e and Supplementary Fig. 24). To study the interactions, we modified the 3′ end of interacting region 1 with the quencher Iowa Black and the 3′ end of interacting region 2 with fluorescein (Fig. 7f, sequence details in Supplementary Fig. 24). We first assembled the DX

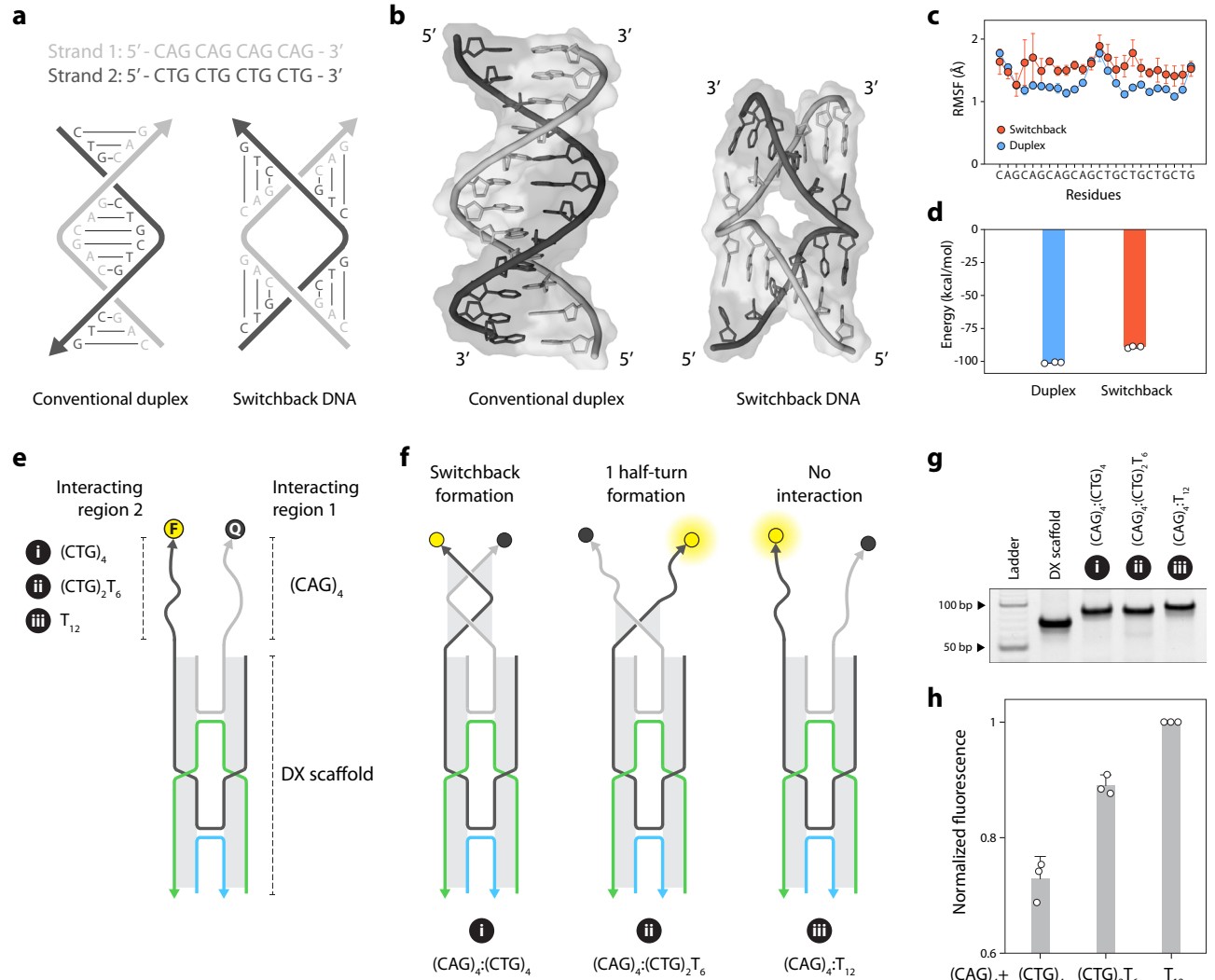

**Fig. 7 | Potential role of switchback DNA in repeat sequences. a** A pair of repeat sequences that can form a conventional duplex or switchback DNA. **b** Simulated structures of repeat sequences forming conventional duplex or switchback DNA. **c** Root mean square fluctuations (RMSF) of residues in conventional duplex and switchback DNA. **d** Calculated thermodynamic parameters of conventional duplex and switchback DNA formed by repeat sequences. **e** Design of a double crossover (DX) motif scaffold to place two interacting regions in parallel orientation. The 3′ ends of the interacting regions contain a fluorophore and a quencher to monitor the interaction. **f** Hybridization of interacting region 1 containing a $(CAG)_4$ sequence with interacting region 2 containing a $(CTG)_4$ sequence forms a switchback DNA,

reducing fluorescence signal. Two controls were used in this study. The interacting region 2 in the first control contains two repeating units of CTG and six thymines (denoted $(CTG)_2T_6$), and the interacting region 2 in the second control contains twelve thymines (denoted $T_{12}$). **g** Non-denaturing PAGE analysis of DX scaffold containing different interacting regions. **h** Fluorescence analysis of the interaction between the two regions on the DX scaffold. $(CAG)_4$:$(CTG)_4$ shows less fluorescence, indicating the formation of a switchback structure. Data represent mean and standard deviations of experiments performed with three replicates. The data shown in (**g**) are representative of experiments performed twice ($n = 2$) with similar reproducible results. Source data are provided as a Source Data file.

scaffold with the different combinations of interacting regions and validated assembly using non-denaturing PAGE and confirmed that the structures are isolated and that there are no higher-order aggregations caused by the single strand extensions on the DX scaffolds (Fig. 7g and Supplementary Fig. 25). We then obtained the fluorescence spectra for each of the annealed structures and quantified the peak fluorescence signals (Fig. 7h). The control structure $(CAG)_4$:$T_{12}$ showed a high fluorescence signal indicating that the two regions do not interact with each other. The fluorescence was slightly reduced (~9%) for the $(CAG)_4$:$(CTG)_2T_6$ pair indicating some interaction between the CAG and CTG domains. The $(CAG)_4$:$(CTG)_4$ version showed a 27% reduction in the fluorescence signal, suggesting that switchback DNA formation quenched the fluorescence by bringing the fluorescein and quencher to close proximity due to base pairing between the two parallel strands. To confirm that the fluorescence quenching is indeed due to sequence-specific base pairing between the parallel strands, we added

a tenfold excess of unlabeled $(CAG)_4$ single strand to the DX scaffold with $(CAG)_4$:$(CTG)_4$ interacting regions. The unlabeled $(CAG)_4$ forms a conventional duplex with the $(CTG)_4$ in interacting region 2, thus displacing the fluorescein-labeled $(CAG)_4$ and restoring the fluorescence (Supplementary Fig. 26). These results provide preliminary data that indicate switchback DNA may form in CAG repeat sequences under certain conditions.

Our hypothesis that switchback DNA could be an alternate secondary structure in repeat sequences is based on the fact that long tracts of repeat sequences are known to adopt several alternate structures[71]. The formation of such alternate structures in repeat sequences is dependent on DNA concentration, repeat length, negative supercoiling stress, and flanking sequences[72,73]. Our experiment using a DX scaffold suggests that under specific conditions, even short $(CAG)_4$:$(CTG)_4$ repeat sequences could form switchback DNA (Fig. 7h). Since the energy required for the formation of most alternative DNA

secondary structures decreases as the repeat length increases[74,75], the potential role of switchback DNA is better considered in longer stretches of repeat sequences rather than the short two-domain structures characterized here. Collectively, our results suggest that switchback DNA could potentially have a biological role in repeat sequences, but further work is needed to validate this hypothesis. The transient and dynamic nature of alternate structures, along with their very low occurrence (0.01–1% of total DNA in some cases[76]) also makes it a challenge to detect these structures[77]. It is of note that PX DNA, another synthetic motif, has been shown to form in homologous segments of superhelical DNA[33] and proteins that structure-specifically bind to PX DNA have been found[35,36].

## Discussion

In this work, we provide a detailed characterization of the structural and dynamic properties of switchback DNA. Our results show that while switchback DNA is stable, it requires higher $Mg^{2+}$ levels compared to a conventional duplex, but within the range of most DNA motifs and nanostructures. By introducing mutations in the sequence, we show domain-dependent stability of switchback DNA, information that is useful in higher-order nanostructure design. The structural preference of component strands to form either switchback DNA or conventional duplex could be used in controlling sub-populations within a mixture[78] and in toehold-less strand displacement[79]. This strategy works on the structure-specific affinity of a set of DNA strands (rather than the typical sequence-based affinity) and will allow for multiple displacement cascades as the displaced strand has a sequence distinct from the complement. Our observation that nuclease resistance levels vary for different structures and that thermal stability (or instability) does not always correspond to biostability is consistent with our earlier work with PX DNA[21]. Our study on small molecule binding provides insight into design-based loading of small molecules on DNA nanostructures, and these results hold significance in studying the drug loading efficiencies of DNA nanostructure-based drug carriers. Our biological studies show that switchback DNA has a marginal effect on cell viability and induces lower immune response than a conventional duplex when selected markers were tested in the HeLa cell line. Specifically, switchback DNA exhibited significantly lower expression of *CXCL-8* compared to duplex. *CXCL-8* has been known to promote tumoral angiogenesis, and reducing the adverse effects of *CXCL-8* signaling may be beneficial in cancer treatment[80]. These results could therefore help inform the design of DNA-based carriers and nanodevices that target the tumor microenvironment. Together, these results provide important information for potential use of the switchback structure in biomedical applications.

In addition to the assembly and characterization of switchback DNA, we explored the implications of the unique modular base pair complementarity paradigm of switchback DNA in a biological context. The sequence requirement for formation of the switchback DNA structure is satisfied by short tandem repeat sequences that are present widely in the eukaryotic genome. Tandem repeats are known to form alternate structures such as hairpins[58], G-quadruplexes[59], triplexes[60], and minidumbbells[81]. While G-quadruplexes are found only in G-rich sequences[59] and minidumbbells are seen in pyrimidine-rich repeat sequences[81], the switchback structure does not have such base composition requirements. Minidumbbells[81] and foldback DNA intercoil[82] that occur in nature require high concentrations of divalent ions to form, indicating that the requirement of higher $Mg^{2+}$ concentrations for switchback DNA does not reduce its biological relevance. Non-B-DNA structures stabilized by $Ca^{2+}$ also occur in dinucleotide $(TG/AC)_n$ repeats, which suggests the possible involvement of divalent ion-stabilized DNA secondary structures in various dynamic processes at the chromosomal level[83]. The secondary structure formation can also be highly dynamic and interchangeable. For example, the purine-rich strand from an H-DNA-forming sequence can fold back to form a triplex structure at

neutral pH in the presence of divalent cations such as $Mg^{2+}$. However, under acidic conditions, the cytosines can be protonated, and the pyrimidine strand can serve as the third strand in the triplex structure[84,85]. We hypothesize that a similar case could exist for two sequences that can hybridize to form either conventional duplex or switchback DNA under specific conditions. We have shown that the primary sequence requirement for the formation of switchback DNA is satisfied by repeat sequences, proximal placement of the complementary strands in parallel orientation may favor its formation, and when folded into switchback form in silico, the repeat sequences can remain stable. However, further investigations are necessary to demonstrate the occurrence of the switchback form in repeat sequences in vivo. We believe that the possible existence of switchback DNA in the genome warrants exploration, for it allows us to understand better the grammar and syntax of the molecular language of nucleic acids in biology. As an alternative DNA conformation that still uses the canonical Watson-Crick-Franklin hydrogen bonding, the switchback structure gains relevance because it allows the expansion of the scope of sequence complementarity. That is, for any given sequence, it is possible to dictate a switchback complement sequence just as readily as the regular complementary sequence. This is significant as it has implications for designing novel sequence-specific nucleic acid interactions and manipulating such interactions in biological systems if they exist in nature.

In DNA nanotechnology, switchback DNA structures can be combined with sticky ends to create double-crossover-like structures that are used in 3D assemblies[41]. Larger constructs based on switchback DNA are possible but may require chemical functionalization (such as 3'–3' or 5'–5' linkages) to allow for strand connectivity since the two strands in the structure are parallel. The left-handedness of switchback DNA can be an advantage in some cases. In topological studies of catenanes and knots, structures constructed using conventional right-handed B-DNA duplexes have negative nodes[86], and require the use of left-handed Z-DNA or L-DNA to produce a positive node. Incorporation of Z-DNA requires specific sequences and ionic conditions while L-DNA requires expensive chemical synthesis. In such cases, the use of a switchback structure will provide a positive node due to its left-handedness. Further, switchback complementarity could be used to connect DNA motifs as an alternate for sticky ended cohesion, such as with bubble-bubble cohesion[87] or paranemic cohesion[88]. The requirement of $Mg^{2+}$ to form stable switchback DNA could also be useful in ion-dependent assembly of DNA nanostructures, similar to DNA nanostructures constructed through loop–loop interactions at high $Mg^{2+}$ concentrations[78]. Overall, our study provides a deeper understanding of the switchback DNA structure with utility in nanotechnology to fold DNA into biostable and functional structures for different applications, and presents a biological perspective for the DNA motif with possible implications in repeat expansion disorders.

## Methods

### Preparation of DNA complexes

DNA oligonucleotides were purchased from Integrated DNA Technologies (IDT) with standard desalting and were used without further purification. Stock solutions of DNA oligonucleotides were prepared in nuclease-free water, and their concentrations were estimated by measuring the absorbance of the DNA solutions at 260 nm using Nanodrop 2000 spectrophotometer (Thermo Scientific) and their respective molar extinction coefficients. To prepare the DNA solution, one or more strands of DNA were taken in the specified molar concentrations in tris-acetate EDTA buffer containing 40 mM Tris base (pH 8.0), 20 mM acetic acid, 2 mM EDTA, and 12.5 mM magnesium acetate (1× TAE-$Mg^{2+}$). For experiments in which $Mg^{2+}$ concentrations were varied, the DNA solution was prepared in 1× TAE without $Mg^{2+}$ and the required amounts of magnesium acetate were added to obtain the desired concentration of $Mg^{2+}$ in the sample. DNA solutions were

annealed in a thermal cycler with the following steps: 95 °C for 5 min, 65 °C for 30 min, 50 °C for 30 min, 37 °C for 30 min, 22 °C for 20 min, and 4 °C for 2 h. The solutions were stored at 4 °C.

## Gel electrophoresis, imaging, and analysis

18% polyacrylamide gels were prepared using 19:1 acrylamide solution (National Diagnostics) in 1× TAE-Mg$^{2+}$ buffer. Annealed DNA complexes were mixed with 1 μl of 10× loading dye containing bromophenol blue and glycerol before loading in the gels. Gels were run at 4 °C in 1× TAE-Mg$^{2+}$ running buffer. Gels were stained with 0.5× GelRed (Biotium) in water for 20 min in the dark and destained in water for 10 min. Imaging was done on a Bio-Rad Gel Doc XR+ imager using the default settings for GelRed with ultraviolet illumination. Images were typically taken at multiple exposures ranging from 5 to 30 s to facilitate accurate quantification. For each gel band, quantification was done using the highest-exposure image that did not contain saturated pixels in the band of interest. For analysis of replicates, three separate gels with the same exposure time were typically used for quantifying band brightness. To quantify each gel band, 12-bit images were imported into ImageJ and processed with a 2-pixel median filter to help reduce noise and small speckles and eliminate hot pixels. A rectangular selection (of common size) was applied to each gel lane. The plot profile command was used to get a mean intensity profile along the bandwidth, and the area under the curve was quantified as a measure of the overall signal.

## Circular dichroism (CD) spectroscopy

Solutions of single-stranded and double-stranded DNA with a strand concentration of 20 μM were prepared in 1× TAE-Mg$^{2+}$ buffer and annealed as described above. 200 μl of the DNA solution was taken in a quartz cuvette of 1 mm thickness. CD spectra were recorded from 200 nm to 360 nm on a Jasco J-815 CD spectrometer with a scan speed of 100 nm/min, bandwidth of 1 nm, and digital integration time was set at 1 s. The data shown are the average of three accumulations.

## Thermal melting studies

UV thermal melting studies were performed on a Cary 3500 UV–Visible Spectrophotometer (Agilent). DNA complexes were annealed at a final strand concentration of 8 μM in 1× TAE buffer containing 12.5 mM magnesium acetate. For experiments in which Mg$^{2+}$ concentrations were varied, DNA samples were prepared in 1× TAE buffer, and the required amounts of magnesium acetate were added, followed by annealing. Absorbance at 260 nm was recorded when the solutions were heated from 20 °C to 90 °C at a rate of 0.5 °C/min. The data were normalized and fitted to the Boltzmann function using OriginPro. The temperature ranges used for fitting the curves were 15–65 °C for switchback DNA and 15–90 °C for the conventional duplexes.

## Isothermal titration calorimetry (ITC)

ITC studies were carried out on Affinity ITC (TA instruments) at 25 °C. DNA solutions were prepared in 1× TAE buffer containing 12.5 mM magnesium acetate (pH 8). The heterodimer switchback and its corresponding conventional duplex were used for ITC. The sample cell was loaded with strand Y (for switchback formation) or strand Z (for duplex formation) (refer to Supplementary Fig. 1) and the reference cell contained deionized water. The syringe was loaded with strand X. A total of 20 injections (2 μl/injection) were made while maintaining a stirring speed of 125 rpm. The heat of dilution was measured by titrating strand X in the syringe against the buffer taken in the cell and subtracted from the sample data. The binding isotherm was fitted to the independent binding model to derive the thermodynamic parameters (see Supplementary Table 2).

## Fluorescence spectroscopy

To measure the fluorescence emission of the DNA-binding small molecules, DNA samples at a constant concentration were mixed with different concentrations of small molecules. For DNA complexes, we used 80 μM nucleotide concentration (we report in nucleotide concentration so that the amount of DNA required for such an experiment can be calculated for structures of other sizes). In our case, both the structures used in this experiment contain 28 nucleotides, thus being at a final concentration of 2.86 μM for the complex. The final concentrations of the small molecules tested were 0 to 60 μM for ethidium bromide and GelRed, 0 to 30 μM for DAPI and 0 to 60 μM for Hoechst 33258 (H33258). Once mixed with the small molecules, the samples were incubated for 2 hours at 20 °C for EBr, GelRed, DAPI, and H33258. To measure fluorescence with EBr and GelRed, samples were placed in a 386-well plate for the emission spectra to be recorded with a PerkinElmer Envision multimode plate reader. For EBr, the excitation wavelength was 520 nm, and the emission at 600 nm was recorded with 900 flashes per well. For GelRed, the excitation wavelength was 272 nm, and the emission at 600 nm was recorded with 500 flashes per well. For samples containing DAPI and H33258, emission spectra were recorded on a HORIBA Jobin Yvon Fluorolog-3-22 spectrofluorometer with a scan speed of 100 nm/min. For DAPI, the excitation wavelength was 360 nm and emission spectra were recorded from 370 to 600 nm. For H33258, the excitation wavelength was 352 nm and emission spectra were recorded from 362 to 600 nm. The enhanced fluorescence at the peak wavelength ($I$–$I_0$) corresponding to each concentration of the small molecules was calculated as the difference between the intensity of emission in the sample with ($I$) and without DNA ($I_0$). The wavelengths used for EBr, GelRed, DAPI, and H33258 were 600, 600, 460, and 460 nm, respectively.

## Fluorescence quenching experiments

Fluorescein (FAM) and Iowa Black (IB) labeled oligonucleotides were purchased from IDT. Heterodimeric switchback DNA XY and conventional duplex XZ were constructed by annealing fluorescein-labeled X-5′FAM and Iowa Black-labeled Y-5′IB or Z-5′IB in 1× TAE-Mg$^{2+}$ at a concentration of 0.5 μM. DX scaffold and DX scaffolds with interacting regions were assembled using the corresponding four strands and annealed in a thermal cycler with the following steps: 90 °C for 3 min, 65 °C for 20 min, 45 °C for 20 min, 37 °C for 30 min, 20 °C for 30 min and stored at 4 °C. Strand combinations for each construct is provided in Supplementary Fig. 24 and sequences are provided in Supplementary Table 1. The fluorescence spectra of the complexes were measured using a PerkinElmer Envision multimode plate reader. The excitation wavelength was 490 nm, and the emission spectra were recorded for a range of 502 nm to 650 nm with 100 flashes. An average of emission recorded at 520 nm for triplicate samples was used to quantify fluorescence quenching.

## Molecular dynamics (MD) simulations and molecular docking

Conventional DNA duplexes were constructed in Molecular Operating Environment (MOE)[89] while the switchback DNA was modeled based on a reference structure from the protein data bank (PDB ID: 8EPB). Using MOE, the sequence of the original structure was mutated to match the sequences used in our experiments. MD simulations were performed using GROMACS 2019.490 and the Chen-Garcia force field. The size of the box and the number of ions and water molecules varied slightly for each construct. On average, the edge of each cubic simulation box was ~7 nm containing 13 Mg$^{2+}$ and 4 Cl$^-$ ions and ~12,137 water molecules. MD simulations incorporated a leap-frog algorithm with a 2-fs time step to integrate the equations of motion. The system was maintained at 300 K using the velocity rescaling thermostat[90] and pressure was maintained at 1 atm using the Berendsen barostat for equilibration[91]. Long-range electrostatic interactions were calculated using the particle mesh Ewald algorithm with a real space cut-off of 1.0 nm[92] and Lennard-Jones electrostatic interactions were truncated at 1 nm. The TIP3P model was used to represent the water molecules, and the LINCS algorithm[93] was used to constrain the motion of

hydrogen atoms bonded to heavy atoms. The system was subjected to energy minimization to prevent any overlap of atoms, followed by a short equilibration (5 ns) and 50-ns production run. Three replicates were run for each system. The simulations were visualized using Visual Molecular Dynamics (VMD) software[94] and analyzed using tools from GROMACS[95]. Hydrogen bonding analysis was performed in VMD using a donor-acceptor distance cutoff of 0.33 nm and hydrogen-donor-acceptor angle cutoff of 30°. All plots were generated in R and structure figures were generated in PyMOL[96].

The thermodynamic properties of conventional duplex and switchback DNA conformations for the CAG repeat sequences were determined using the MMPBSA method[65] (Supplementary Table 5). One of the DNA strands was designated as the 'receptor' and the other as the 'ligand'. The enthalpic contributions were derived from the sum of molecular mechanics ($\Delta E_{MM}$) and solvation free energy ($\Delta G_{SOL}$). The molecular mechanics component accounts for changes in bonds, angles, dihedrals, electrostatics ($\Delta E_{EL}$), and van der Waals ($\Delta E_{VDW}$) interactions as a result of receptor-ligand interaction. Since there is no change in bonded interactions when the two strands designated as receptor and ligand come together to form a duplex or a switchback, they do not contribute to $\Delta E_{MM}$. The solvation free energy comprises of polar ($\Delta G_{PB}$) and non-polar ($\Delta G_{NP}$) components determined by Poisson Boltzmann (PB) model and the solvent accessible surface area (SA) model. The input parameters for these calculations were chosen based on a prior study that successfully correlated duplex energies with experimentally observed values[97]. We used 25,000 snapshots from the 50 ns MD simulations, the bsc1 forcefield[98] for DNA and the TIP3P water model[99] for the calculations to align with the simulation parameters.

Molecular docking was performed using MOE. Starting with the relaxed structure of the duplex or the switchback DNA, we used sequential docking to dock one small molecule at a time until the number of contacts between the small molecule and the DNA was less than two in a docking run. While groove-binding molecules (e.g.: Hoechst 33258) do not distort the structure of the DNA, intercalators (eg: ethidium bromide) require the disruption of stacking interactions to accommodate the small molecules. Since we used only localized flexibility during docking, such larger conformational changes cannot be accommodated during docking. To overcome this limitation, we created potential intercalation sites on the duplex and switchback before docking. For each docking run, 20 initial poses and 3 refined poses were generated. The top prediction was then chosen for the next round of docking. We used the default docking parameters in MOE to set up each docking run.

### Biostability assay

Annealed DNA complexes (at 1 μM) were first mixed with reaction buffers and other components provided with the enzymes (final of 1×). Enzyme dilutions were made in nuclease-free water. For the nuclease degradation assay, 1 μl of the enzyme was added to 10 μl of the sample containing DNA complex and reaction buffer (and any other required components prescribed for each enzyme by the vendor, see Supplementary Table 4). Typically, samples were incubated at 37 °C for 1 h for the enzyme concentration series (Fig. 5b). For testing nuclease degradation at different time points (Fig. 5c), the enzyme was added to DNA/reaction buffer solution at different time intervals, starting with the longest time point, and samples were loaded quickly on to gels at time 0 for gel analysis. Incubated samples were mixed with gel loading dye containing bromophenol blue and 1× TAE-Mg²⁺ and run on non-denaturing gels to analyze degradation over time[53]. Gels were stained with 0.5× GelRed (Biotium) in water for 20 min in dark, destained in water for 10 min and imaged on a Bio-Rad Gel Doc XR+ imager using the default settings for GelRed with UV illumination and analyzed using Image J.

### Cell culture and treatments

HeLa cells were cultured in Dulbecco's modified Eagle medium supplemented with 10% fetal bovine serum and 1% penicillin/streptomycin (P/S) under standard conditions of 37 °C and 5% $CO_2$. The cells were seeded into 12-well cell culture plates at a seeding density of $2.5 \times 10^4$ cells per well. Once the cells reached ~70% confluency, they were washed with PBS, and provided with fresh growing media containing switchback DNA and duplex DNA at 100 nM, 250 nM, 500 nM, and 1000 nM in 1×TAE-Mg²⁺ and buffer (without DNA) as a control. Cells were also treated with polyinosinic-polycytidylic acid potassium salt (polyIC, SigmaMillipore, P9582) at 1 μg/ml, 10 μg/ml, and 100 μg/ml in 1× PBS with PBS buffer as a control. Cells were harvested after 24 h of treatment.

### Cytotoxicity assay

Cytotoxicity of switchback DNA, conventional duplex, and polyIC incubation was evaluated using the MTT (3-(4,5-dimethylthiazol-2-yl) −2,5-diphenyltetrazolium bromide dye) assay (Sigma Aldrich). The cells were seeded into 96 well-format cell culture plates at a seeding density of $8 \times 10^3$ cells per well. After 24 h of treatment with switchback DNA, conventional duplex, or polyIC in triplicates at the concentrations indicated above, cells were washed with PBS and replaced with a 1:1 ratio of fresh growth medium and 5 mg/ml of MTT (Sigma Aldrich) stock solution prepared in 1× PBS and incubated for 4 h at 37 °C in 5% $CO_2$ to allow the intracellular reduction of the soluble yellow MTT to insoluble formazan crystals. The formazan crystals were solubilized by the addition of 100 μl of DMSO at room temperature and shaking on an orbital shaker for 5 min Absorbance was measured at 590 nm on a microplate reader (Synergy H1 multi-mode microplate reader). The percentage of living cells was calculated by using the following equation where λ590 is absorbance measured at 590 nm. As controls, we used cells incubated with 1× TAE-Mg²⁺ (buffer used to assemble the DNA structures) and normalized the values to this result.

$$\text{Cell viability}\,(\%) = \lambda590\,(untreated\ cells - treated\ cells)\,/\lambda590(untreated\ cells) \times 100 \quad (1)$$

### RT-qPCR analysis

Total cellular RNA was isolated using Quick-RNA Midiprep Kit (Zymo Research Corporation) with on-column DNase I treatment at room temperature for 15−20 min. RNA concentration was measured using NanoDrop (Thermo Scientific) and a total of 200 ng of total RNA was used for cDNA synthesis in a 20 μl reaction with the SuperScript IV kit (Thermo Fisher) and random hexamer primers (IDT). 2 μl of cDNA was used to perform RT-qPCR using specific primer sets listed in Supplementary Table 6.

### Reporting summary

Further information on research design is available in the Nature Portfolio Reporting Summary linked to this article.

## Data availability

The data that support the findings of this study are available within the paper and its supplementary information files, including uncropped gel images of representative replicates for all experiments. Source data are provided with this paper.

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

## Acknowledgements

Research reported in this publication was supported by the National Institutes of Health (NIH) through the National Institute of General Medical Sciences (NIGMS) under award number R35GM150672 to A.R.C., State University of New York institutional start-up funds to K.R. and NSF SAGES grant to S.V. J.M.L. was supported by a Myotonic Dystrophy Foundation Postdoctoral Fellowship. We thank Ken Halvorsen and Jibin Abraham Punnoose for critical discussions on the project and comments on the manuscript.

## Author contributions

B.R.M. and A.R.C. designed experiments. B.R.M. performed ITC, CD, and UV melting experiments and analyzed the corresponding data. B.R.M., H.T., and A.R. performed gel electrophoresis for assembly and characterization of DNA complexes. B.R.M. and A.R. performed fluorescence spectroscopy experiments and analyzed the corresponding data. H.T., A.R., and A.R.C. performed gel electrophoresis for biostability experiments. B.R.M. and A.R.C. analyzed gel electrophoresis results. H.Z. and S.V. performed molecular dynamics simulations and molecular docking and analyzed the corresponding data. J.M.L. performed cell studies and J.M.L. and K.R. analyzed the corresponding data. A.R.C. conceived and supervised the project and visualized the data. B.R.M. and A.R.C. wrote the manuscript with contributions from S.V. and K.R.

## Competing interests

The authors have no competing interests.
