## [Peer Review File · Nature Communications]

Reviewers' Comments:

Reviewer #1:

Remarks to the Author:

The manuscript presents an impressively wide range of experiments on the non-canonical DNA form referred to as a 'switchback' structure. The work is noteworthy because it lays the foundations for the use of switchback DNA for nanotechnology applications and also postulates a biological role for this structure. Overall this is a good piece of work, and the manuscript is well-written. The study opens up several avenues for further research, as aspects of the investigation could be probed in more depth.

I would support the publication of this work after appropriate revisions, as listed below. To summarise, I think the authors need to address some important points regarding the results, add a reference and clarify a couple of very minor points on the methods. I note that the manuscript is currently presented in such a way that it is likely to appeal most strongly to researchers in DNA nanotechnology but it could potentially be revised to increase appeal to a wider audience.

- Figure 1, thermodynamics and mismatches: given the significance of the base sequence in determining thermodynamics of hybridisation in DNA duplexes, I think the authors need to say/do more to convince the reader that their results would generalise to other sequences. In principle, this could be achieved with further experimentation, but that is not essential at this stage because a full investigation of the effect of all sequence variations would probably constitute an entire paper in its own right. Unless the authors do wish to include data from further experiments, I would recommend that they insert additional text elaborating on reasons behind the choice of sequences, the GC content, the nature of the mismatches and why sequence-specific effects would not undermine the conclusions.
- For researchers wishing to design their own switchback structures, it would be helpful if the authors could include (maybe in the SI) a brief description of how the strands can be designed. This may be covered in part by the point above (reasons behind the choice of sequences).
- The authors state 'for the switchback structure, presence of one mismatch did not affect the assembly.' The data seems to show otherwise. Fig. 1i shows two bands in the switchback-1mm case. Fig 1j and 1k also show a shift in melting temperature. Can the authors comment and/or amend the manuscript?
- Strand displacement experiments (Fig 2c and 2d): I was not entirely clear on whether there is a toehold here – can the authors please clarify? Also, can the authors comment on the different lengths of the constructs? How significant is the number of base pairs? (It feels as if there is another follow-up paper here, this time on the phenomenon of strand displacement in switchback DNA, investigating all possible variables.)
- Fig 5 – the micrographs are too small.
- Discussion on relevance of switchback DNA in tandem repeat sequences: this needs some expansion. It feels too brief for a hypothesis of this significance. I would also like to see some clarification on the sentence 'the overall structure of the switchback DNA is well tolerated by the repeat sequences' – earlier in the paper the authors have argued that DNA prefers to be in duplex form than in switchback form so they need to elaborate on how the switchback form could be stable in vivo.
- Introduction section: I think the switchback structure is the same as or similar to the crossover stack depicted in Fig. 4a of the 2012 PNAS paper by Bai et al (<https://www.pnas.org/doi/full/10.1073/pnas.1215713109>). If I'm right, I think this reference should be added, together with a sentence of comment.
- Methodology appears sound and the methods are described well. I think some minor methodological clarifications are required:
 - o For gels, they state that analysis is performed with ImageJ. I think a little more information on quantification of band intensity is required.
 - o For fluorescence spectroscopy, the authors say '80uM (nucleotide concentration) DNA solutions were mixed with...' Can the authors confirm if 80uM is the stock concentration or the final

concentration? I think they should also unpack the phrase 'nucleotide concentration' to avoid confusion.

- Optional: The authors may wish to consider further the target audience for their paper. The structure of the manuscript suggests that it is primarily aimed at researchers in the DNA nanotechnology arena, with the potential biological role of switchback DNA being an interesting addendum. However, if the authors are correct in their hypothesis about the role of switchback DNA in disease states, their work is relevant to a wider audience. If the authors wish to reach this audience, I would recommend rethinking the abstract, opening paragraph and perhaps even the title.
- Optional: the authors could mention in the discussion of Fig. 1 that the switchback DNA does not stain as strongly as the ds complexes, paving the way for the later work with small molecule binding.

Katherine Dunn, 29th January 2024

Reviewer #2:

Remarks to the Author:

In this manuscript, "Unusual Structural Properties of Switchback DNA", the authors present a detailed characterization of switchback DNA compared to canonical B-DNA duplexes. This synthetic DNA motif, discovered many years ago, is of interest due to its potential use in nucleic acid nanotechnology applications. The authors used a variety of techniques (in silico and in vitro) to characterize properties of switchback (relative to B-DNA duplex substrates) such as thermal stability, biostability (as assessed by digestion by several nucleases), small molecule binding experiments, cell viability, and immune responses. Their results indicate that there are some interesting differences in these properties between switchback DNA and B-DNA duplexes.

This is a timely study that has the potential to have a positive impact in guiding the development of new applications for synthetic DNA motifs in nanotechnology. The authors have provided a variety of approaches to draw their conclusions. However, some of the conclusions drawn are not completely convincing with the data as currently presented. Below are listed several major concerns to be addressed by the authors.

Major concerns to be addressed include:

1. While this is a detailed characterization of switchback DNA, the authors may want to better explain the novelty and potential impact of the work on the field. For example, the work in cells is minimal and the computational simulation and prediction studies to suggest switchback DNA forms a biologically relevant alternative structure involved in repeat-expansion diseases is not convincing without data providing some type of biological evidence (see comment #7 below).
2. The data presented in Figure 1 are a bit confusing and/or not completely convincing. The authors state that the switchback DNA migrated in non-denaturing polyacrylamide gels in a position similar to that of the B-DNA duplex. So how is this evidence for structure formation? They then designed a longer duplex complement so that the two substrates would migrate differently due to their lengths but again...how does this distinguish between the 2 structures rather than just the 2 sequences? The authors need to clarify their rationale and interpretation of the data and/or provide a different approach to better demonstrate the formation of switchback DNA to strengthen their conclusions.
3. Related to comment #2 above, in Fig. S6, the authors state the CD signature of switchback DNA was similar to that of the B-DNA duplex, "confirming that the half-turns in the switchback DNA resemble a typical B-DNA structure despite the overall left-handed nature of the switchback structure". This is not solidly convincing because if the CD signatures are the same, then what evidence is there to support the formation of the switchback structure under the conditions of the

assay? The authors should provide a better rationale for this experiment and/or provide additional evidence using an orthogonal assay, which would strengthen the conclusions drawn in this study.

4. The mismatch tolerance experiments using UV melting (Fig. 1k) are more convincing than the those using PAGE (Fig. 1i); however, the results from the 2 different types of experiments seem to differ in regard to the stability of the duplex DNA containing 2 mismatches. Could the authors comment on this apparent discrepancy?

5. In Fig. 4, it is not clear how the experiments were performed, but it seems to be done using purified enzymes and purified DNA substrates in vitro. Were the buffer conditions physiologically relevant? And while it did appear that the different nucleases tested had varying activity levels on the switchback vs. duplex substrate, would these differences still hold in cells? Would this make a difference regarding the use of switchback DNA in nanotechnology-based applications? The authors should discuss the relevance of these in vitro results. Further, if the authors demonstrated the relative stability of the switchback substrates to duplexes in cells, this would strengthen the impact of this study.

6. In the cell viability and immune response experiments in HeLa cells (Fig. 5), the DNA substrates were incubated for 24-48 hours. Were the DNA substrates still intact after that time in cells? What was the percentage of DNA substrate remaining? The authors should determine the half-life of the various DNA substrates in cells. This would be informative and would strengthen the results on cell viability and immune responses shown.

7. The experiments on the "biological significance" (Fig. 5) of sequences that can form switchback DNA are not very convincing as presented. First, the authors show a preference for duplex formation over switchback DNA in Fig. 2, so what is the likelihood that these sequences would form switchback DNA in cells? The results shown are done using molecular dynamics and simulation studies. While this suggests that these structures may form in cells (and therefore, may have biological significance), the results are not convincing without some cell-based data with a biological outcome. Second, to suggest that switchback DNA may be involved in repeat-mediated diseases is quite speculative and premature. In order to be convincing, the authors would have to test the switchback substrates in comparison to the number of other alternative DNA structures that have been strongly implicated in repeat-mediated diseases in studies to determine structure formation, mutagenic potential, etc., etc.

Response to reviewers

The unusual structural properties and potential biological relevance of switchback DNA

Dear Reviewers,

Thank you for taking the time and effort to review our manuscript. We appreciate your comments and constructive criticisms and have addressed your concerns to the best of our abilities. We have performed additional experiments and revised the text to improve the manuscript. We are excited for this opportunity to publish our work in *Nature Communications*.

Reviewer 1:

The manuscript presents an impressively wide range of experiments on the non-canonical DNA form referred to as a 'switchback' structure. The work is noteworthy because it lays the foundations for the use of switchback DNA for nanotechnology applications and also postulates a biological role for this structure. Overall this is a good piece of work, and the manuscript is well-written. The study opens up several avenues for further research, as aspects of the investigation could be probed in more depth.

I would support the publication of this work after appropriate revisions, as listed below. To summarise, I think the authors need to address some important points regarding the results, add a reference and clarify a couple of very minor points on the methods. I note that the manuscript is currently presented in such a way that it is likely to appeal most strongly to researchers in DNA nanotechnology but it could potentially be revised to increase appeal to a wider audience.

Thank you for noting the wide range of experiments performed and indicating that this is a good piece of work and well-written.

1) Figure 1, thermodynamics and mismatches: given the significance of the base sequence in determining thermodynamics of hybridisation in DNA duplexes, I think the authors need to say/do more to convince the reader that their results would generalise to other sequences. In principle, this could be achieved with further experimentation, but that is not essential at this stage because a full investigation of the effect of all sequence variations would probably constitute an entire paper in its own right. Unless the authors do wish to include data from further experiments, I would recommend that they insert additional text elaborating on reasons behind the choice of sequences, the GC content, the nature of the mismatches and why sequence-specific effects would not undermine the conclusions.

In theory, the study and results presented here should be generalizable to other sequences with the complementarity pattern required to form the switchback DNA structure. For our study, we based our design on the report by Chengde Mao's group (ref 41 in manuscript). We would also like to note that another recent study by Chengde Mao's group used several homo- and hetero-complexes of the switchback structure with different sequences as part of a "DX-like (DXL) motif" (ref 42 in manuscript). Together, our study and these prior reports have used different switchback DNA sequences containing different GC contents (shown in Supporting Figure 1 below).

We've revised the text to elaborate on the reasons behind the choice of sequences, the GC content and the generalizability of the results. We've also included additional text on the choice of mismatch location and its effect on the stability. New text and results are provided below for convenient reference:

In introduction:

*"The sequence we used for the heterodimeric switchback DNA is derived from a DX-like motif reported recently,⁴² with the interacting regions containing 67% GC content. Based on our assembly of the homodimeric and heterodimeric switchback DNA as well as a recently reported crystal assembly, other sequences could also be designed for assembling a switchback DNA structure based on a modular base pairing scheme (**Supplementary Fig. 6**) and containing different GC contents. Switchback DNA with 3 and 4 half-turn domains have also been created but increasing the number of half-turns further may induce aggregation of the structures."⁴¹*

In mismatch tolerance section:

"In these examples, we chose the location of the mismatches to be in the middle of the half-turn domains, away from the point where the strands switch back into the next half-turn domain, and note that the specific location of the mismatch within the half-turn domains may have different effects on the stability of the structure."

Supporting Figure 1. GC content of experimentally used switchback sequences.

2) For researchers wishing to design their own switchback structures, it would be helpful if the authors could include (maybe in the SI) a brief description of how the strands can be designed. This may be covered in part by the point above (reasons behind the choice of sequences).

In addition to the details provided in response to the above comment, we have also now provided a scheme to generalize the sequence design for a switchback DNA structure (new Supplementary Fig. 6) and added relevant text in the revised manuscript.

*“Based on our assembly of the homodimeric and heterodimeric switchback DNA as well as a recently reported crystal assembly, other sequences could also be designed for assembling a switchback DNA structure based on a modular base pairing scheme (**Supplementary Fig. 6**) and containing different GC contents. Switchback DNA with 3 and 4 half-turn domains have also been created but increasing the number of half-turns further may induce aggregation of the structures.⁴¹”*

3) The authors state ‘for the switchback structure, presence of one mismatch did not affect the assembly.’ The data seems to show otherwise. Fig. 1i shows two bands in the switchback-1mm case. Fig 1j and 1k also show a shift in melting temperature. Can the authors comment and/or amend the manuscript?

When one mismatch is present in the sequence, we observe a prominent band close to the position of the duplex, and this band is also different from the bands corresponding to single strands (new Supplementary Fig. 10). The band seems to migrate slightly more on the gel compared to the one with no mismatch. This could be due to the presence of mismatch in the duplex. When two mismatches are present, there is no band corresponding to a 2-strand complex, indicating that it is almost completely broken down into ssDNA at equilibrium. Further characterization with UV-melting study showed that the melting temperature of the structures were decreased by 5 °C when one mismatch was introduced. The difference in stability of the structures with two adjacent or two separated mismatches was evident in the UV-T_m results which showed that the T_m decreased by 16 °C and 8 °C, respectively.

We’ve also revised the text accordingly:

*“Non-denaturing PAGE analysis showed that the formation of conventional duplex was not affected by the presence of one or two mismatches, as indicated by the appearance of the band similar to the control structure without any mismatches (**Fig. 2k**, lanes 2-5). For the switchback structure, the presence of one mismatch slightly affected the assembly. We observed a band corresponding to the switchback DNA with a single mismatch (lane 7) that migrated closer to the position of the structure without any mismatches (lane 6) and is different from the band corresponding to single strands, indicating that the structure is still predominantly intact (**Supplementary Fig. 10**).”*

Supplementary Fig. 10. Non-denaturing polyacrylamide gel image of conventional duplex and switchback DNA with mismatched base pairs.

4) Strand displacement experiments (Fig 2c and 2d): I was not entirely clear on whether there is a toehold here – can the authors please clarify? Also, can the authors comment on the different lengths of the constructs? How significant is the number of base pairs? (It feels as if there is another follow-up paper here, this time on the phenomenon of strand displacement in switchback DNA, investigating all possible variables.)

The sequences do not have any toeholds. The displacement-based conversion between switchback DNA and conventional duplex is entirely “structure-based” as opposed to the typical toehold-based strand displacement that is “sequence-based”.

It is possible to extend the switchback structure to contain more than two half-turn domains. Earlier work by Mao et al, showed that switchback structure remains stable when the number of half-turn domains are increased to up to four. Further increase in length could lead to formation of higher order aggregates.

We have now added additional text to clarify both these points. New text is provided below:

“These results indicate that a duplex complement can displace a switchback complement from a pre-assembled switchback DNA structure, a phenomenon that is essentially toehold-less DNA strand displacement based on structural stability and not just sequence affinity.”

5) Fig 5 – the micrographs are too small.

We've now revised the figure to include zoomed-in micrographs (Fig. 6 in the revised manuscript).

6) Discussion on relevance of switchback DNA in tandem repeat sequences: this needs some expansion. It feels too brief for a hypothesis of this significance. I would also like to see some clarification on the sentence 'the overall structure of the switchback DNA is well tolerated by the repeat sequences' – earlier in the paper the authors have argued that DNA prefers to be in duplex form than in switchback form so they need to elaborate on how the switchback form could be stable in vivo.

We were also asked a similar question by Reviewer 2. We've addressed this both by preliminary experiments and additional MD simulations as well as additional justification based on prior literature.

In brief:

1) We designed a double-crossover (DX) motif-based scaffold to place two adjacent repeat sequences in parallel orientation and show that they form switchback structure using fluorescence experiments (more details below in revised text and new Fig. 7e-h).

2) We've added more text to discuss how switchback DNA could potentially have a biological role despite the low melting temperature and preference towards conventional duplexes.

The revised text describing the new experiments and hypothesis are provided below:

Additional MD analysis:

*"Additionally, we used the Molecular Mechanics/Poisson Boltzmann Surface Area (MMPBSA) method⁶⁶ to calculate thermodynamic properties of the two configurations within the repeat sequence, from the simulation trajectories. The predicted ΔH difference between the duplex and the switchback conformations is ~ 11 kcal/mol (**Fig. 7d**) and primarily arises from altered electrostatics and solvation between the two conformations (**Supplementary Table 5**). This indicates that both conformations are energetically feasible even if the duplex conformation is slightly more favorable."*

New experiment with repeat sequences:

*"Next, we sought to test whether CAG/CTG repeats could adopt switchback conformation. Since the formation of antiparallel conventional duplex is preferred over parallel switchback DNA for the two complementary strands, we imposed geometric constraints on the strands by fixing them on a DNA double crossover (DX) scaffold that places the two sequences next to each other in the parallel orientation (**Fig. 7e**). The DX motif contains two adjacent double helical domains connected by two crossover points, a motif well characterized in our group and several others.^{8,67-71} We extended one*

strand from each double helical domain to create single stranded “interacting regions”, where both extensions are 5’ to 3’ from the DX scaffold. The design also contains a TT linker between the DX scaffold and the interacting regions to allow some flexibility. We fixed interacting region 1 to contain (CAG)₄ and varied the sequences in interacting region 2 to contain (i) (CTG)₄ that can form a switchback with (CAG)₄ in interacting region 1, (ii) (CTG)₂T₆ that can form only one domain of the switchback DNA or (iii) a T₁₂ sequence that cannot recognize (CAG)₄ (**Fig. 7e** and **Supplementary Fig. 24**). To study the interactions, we modified the 3’ end of interacting region 1 with the quencher Iowa Black and the 3’ end of interacting region 2 with fluorescein (**Fig. 7f**, sequence details in **Supplementary Fig. 24**). We first assembled the DX scaffold with the different combinations of interacting regions and validated assembly using non-denaturing PAGE and confirmed that the structures are isolated and that there are no higher-order aggregations caused by the single strand extensions on the DX scaffolds (**Fig. 7g**). We then obtained the fluorescence spectra for each of the annealed structures and quantified the peak fluorescence signals (**Fig. 7h**). The control structure (CAG)₄:T₁₂ showed a high fluorescence signal indicating that the two regions do not interact with each other. The fluorescence was slightly reduced (~9%) for the (CAG)₄:(CTG)₂T₆ pair indicating some interaction between the CAG and CTG domains. The (CAG)₄:(CTG)₄ version showed a 27% reduction in the fluorescence signal suggesting that switchback DNA formation quenched the fluorescence by bringing the fluorescein and quencher to close proximity due to base pairing between the two parallel strands. To confirm that the fluorescence quenching is indeed due to sequence-specific base pairing between the parallel strands, we added a 10-fold excess of unlabeled (CAG)₄ single strand. The unlabeled (CAG)₄ forms a conventional duplex with the (CTG)₄ in interacting region 2, thus displacing the fluorescein-labeled (CAG)₄ and restoring the fluorescence (**Supplementary Fig. 25**). These results provide preliminary data that indicate switchback DNA may form in repeat sequences under certain conditions. Alternate secondary or tertiary structures observed in repeat sequences generally occur in long stretches, are influenced by flanking sequences and can be transient.^{72,73} The DX scaffold is intended to provide this structural context for the two strands to interact. Based on these results and an analysis of direct repeat sequences, we hypothesize that short tandem repeat sequences with repeat lengths of 2, 3, 4, 6, and 12-nt may adopt switchback DNA as well as conventional duplex (**Supplementary Fig. 22**) and an equilibrium between the structures could have implications for repeat expansion diseases.”

Fig. 7. Potential role of switchback DNA in repeat sequences. (a) A pair of repeat sequences that can form a conventional duplex or switchback DNA. (b) Simulated structures of repeat sequences forming conventional duplex or switchback DNA. (c) Root mean square fluctuations (RMSF) of residues in the conventional duplex and switchback. (d) Calculated thermodynamic parameters of the conventional duplex and switchback DNA formed by repeat sequences. (e) Design of a double crossover (DX) motif scaffold to place two interacting regions in parallel orientation. The 3' ends of the interacting regions contain a fluorophore and a quencher to monitor interaction. (f) Hybridization of interacting region 1 containing a (CAG)₄ sequence with interacting region 2 containing a (CTG)₄ sequence forms a switchback DNA, reducing fluorescence signal. Two controls were used in this study. The interacting region 2 in the first control contains two repeating units of CTG and six thymines (denoted (CTG)₂T₆) and the interacting region 2 in the second control contains twelve thymines (denoted T₁₂). (g) Non-denaturing PAGE analysis of DX scaffold containing different interacting regions. (h) Fluorescence analysis of the interaction between two regions on the DX scaffold. (CAG)₄:(CTG)₄ shows less fluorescence indicating formation of a switchback structure. Data represent mean and standard deviations of experiments performed with three replicates.

Additional text revisions for biological relevance (in biological relevance section):

“The modular base pairing scheme of switchback DNA is distinct from the conventional end-to-end base pairing model. While molecular recognition between nucleic acids mainly occurs via the

conventional base pairing model, it is conceivable that the switchback DNA mode of interaction may also exist in certain structural contexts between pairs of sequences that allow such pairing. We hypothesize that switchback DNA could be a potential alternate structure in short tandem repeat sequences. That is, a pair of sense and antisense strands containing tandem direct repeat sequences could form either the switchback DNA or a conventional duplex (**Fig. 7a**). We screened various direct repeat sequences involved in repeat expansion diseases and observed that in some cases, a given sequence could have the same switchback complement and duplex complement (**Supplementary Fig. 22**)."

...

"Our hypothesis that switchback DNA could be an alternate secondary structure in repeat sequences is based on the fact that long tracts of repeat sequences are known to adopt several alternate structures.⁷⁴ The formation of such alternate structures in repeat sequences is dependent on DNA concentration, repeat length, negative supercoiling stress and flanking sequences.^{72,73} Our experiment using a DX scaffold suggests that under specific conditions, even short (CAG)₄:(CTG)₄ repeat sequences could form switchback DNA (**Fig. 7h**). Since the energy required for the formation of most alternative DNA secondary structures decreases as the repeat length increases,^{75,76} the propensity to form switchback DNA may increase with increasing number of repeats. Thus, switchback DNA could occur in long tracts of repeat sequences in the conformationally dynamic genomic DNA although conventional duplex is the preferred form in free oligonucleotides with two half-turn domains. Collectively, this information suggests that switchback DNA could potentially have a biological role in repeat sequences, but further work is needed to validate this hypothesis. The transient and dynamic nature of alternate structures, along with their very low occurrence (0.01 to 1% of total DNA in some cases⁷⁷) also makes it a challenge to detect these structures.⁷⁸ It is of note that PX DNA, another synthetic motif, has been shown to form in homologous segments of superhelical DNA³⁴ and proteins that structure-specifically bind to PX DNA have been found.^{36,37}"

In discussion:

"In addition to the assembly and characterization of switchback DNA, we explored the implications of the unique modular base pair complementarity paradigm of switchback DNA in a biological context. It is known that under certain conditions, long tracts of repeat sequences tend to adopt alternate secondary structures.⁷⁴ The sequence requirement for formation of the switchback DNA structure is satisfied by short tandem repeat sequences that are present widely in the eukaryotic genome.

...

The secondary structure formation can also be highly dynamic and interchangeable. For example, the purine-rich strand from an H-DNA-forming sequence can fold back to form a triplex structure at neutral pH in the presence of divalent cations such as Mg²⁺. However, under acidic conditions, the cytosines can be protonated, and the pyrimidine strand can serve as the third strand in the triplex structure.^{85,86} A similar case could exist for two sequences that can hybridize to form either conventional duplex or switchback DNA under specific conditions."

7) Introduction section: I think the switchback structure is the same as or similar to the crossover stack depicted in Fig. 4a of the 2012 PNAS paper by Bai et al (<https://www.pnas.org/doi/full/10.1073/pnas.1215713109>). If I'm right, I think this reference should be added, together with a sentence of comment.

The citation you mentioned discusses the layered double helical domains in a DNA origami structure. The crossover stacks in Fig 4a do not comprise a switchback DNA double helix. Using the below Supporting Figure 2, we clarify the relation between switchback DNA and multi-helical domain DNA objects.

The switchback DNA structure contains two complete (or continuous) strands that form a globally left-handed helix while the half-turns still contain right-handed B-DNA form (**Supporting Figure 2a**). In DNA motifs contain more than one double helical domain, such as the double crossover (DX) motif, one could imagine a partial switchback where one strand is complete and the other is fragmented (**Supporting Figure 2b**, Rodriguez et al, *Small* 2023, 19, 2300040).

(1) The paper you indicated describes a similar scenario but in a DNA origami structure that contains multiple stacked double helical domains (**Supporting Figure 2c**, Bai et al, *PNAS* 2012, 109, 20012). The authors of this work describe the structure to contain "a synthetic pseudohelix in which one strand follows a left-handed helical path with base pairs pointing along the helical direction, rather than orthogonally to the helical axis as in B-form dsDNA" (**Supporting Figure 2d**). However, this portion of the DNA origami contains only one strand that runs across the helical domains and other strands are connected elsewhere in the structure. Thus, this structure does not form a complete switchback double helical structure. The cartoon on the right of Supporting Figure 2d shows how a full switchback DNA in this case would have looked like.

(2) In addition, such an occurrence is observed in many planar multi-helical domain DNA nanostructure reported in other earlier studies (including the DX and DNA origami mentioned above). For example, in a double-double crossover (DDX) motif, one can see a similar strand that is woven in a left-handed fashion across four double helical domains while the complementary portions are disconnected (**Supporting Figure 2e**, Reishus et al, *JACS* 2005, 127, 17590). These are all instances where only one strand runs through and do not contain a continuous switchback complement to form switchback DNA.

Supporting Figure 2. Comparison of switchback DNA with existing features of other DNA nanostructures.

8) Methodology appears sound and the methods are described well. I think some minor methodological clarifications are required:

-- For gels, they state that analysis is performed with ImageJ. I think a little more information on quantification of band intensity is required.

We've expanded on the gel analysis in the methods section. Revised version is provided below for reference:

"Imaging was done on a Bio-Rad Gel Doc XR+ imager using the default settings for GelRed with ultraviolet illumination. Images were typically taken at multiple exposures ranging from 5 to 30 s to

facilitate accurate quantification. For each gel band, quantification was done using the highest-exposure image that did not contain saturated pixels in the band of interest. For analysis of replicates, three separate gels with the same exposure time were typically used for quantifying band brightness. To quantify each gel band, 12-bit images were imported into ImageJ and processed with a 2-pixel median filter to help reduce noise and small speckles and to eliminate hot pixels. A rectangular selection (of common size) was applied to each gel lane. The plot profile command was used to get a mean intensity profile along the band width, and the area under the curve was quantified as a measure of the overall signal."

-- For fluorescence spectroscopy, the authors say '80uM (nucleotide concentration) DNA solutions were mixed with...' Can the authors confirm if 80uM is the stock concentration or the final concentration? I think they should also unpack the phrase 'nucleotide concentration' to avoid confusion.

We reported DNA concentration in nucleotides so that the concentration can be generalizable for structures of any size. In this case, both the DNA complexes used in the fluorescence experiments contain 28 nucleotides and were used at 2.86 uM final concentration (i.e. 80 uM nucleotide conc). Thus, for larger DNA complexes, the required DNA concentration of the structure would be lower if scaled to the same 80 uM nucleotide concentration. We've revised the text to elaborate these details:

"To measure the fluorescence emission of the DNA-binding small molecules, DNA samples at a constant concentration were mixed with different concentrations of small molecules. For DNA complexes, we used 80 μ M nucleotide concentration (we report in nucleotide concentration so the amount of DNA required for such an experiment can be calculated for structures of other sizes). In our case, both the structures used in this experiment contain 28 nucleotides, thus being at a final concentration of 2.86 μ M for the complex..."

9) Optional: The authors may wish to consider further the target audience for their paper. The structure of the manuscript suggests that it is primarily aimed at researchers in the DNA nanotechnology arena, with the potential biological role of switchback DNA being an interesting addendum. However, if the authors are correct in their hypothesis about the role of switchback DNA in disease states, their work is relevant to a wider audience. If the authors wish to reach this audience, I would recommend rethinking the abstract, opening paragraph and perhaps even the title.

Thank you for this suggestion. In addition to the in vitro experiment showing switchback DNA formation by repeat sequences, we have now rephrased the title and introduction to widen our target audience for the article. Additional experiments and revised text on biological relevance are provided in above points. The additional introductory paragraph is provided below:

*"The motifs used in the field of DNA nanotechnology were initially inspired by structures that exist in nature.²⁹ For example, immobile 4-way junctions were inspired by Holliday junctions.³⁰ Double crossover molecules^{8,31} have been proposed as intermediates in recombination,³² and have been demonstrated to be involved in meiosis.³³ While naturally occurring motifs can guide the design of DNA nanostructures, studying the properties of basic DNA motifs could provide insights into biological phenomena as well. There are several examples for this. PX DNA has been suggested to be involved in interactions between homologous segments of supercoiled DNA in processes such as recombination and repair.^{34,35} Further, proteins that bind to PX DNA in a structure-dependent fashion have been isolated,³⁶ and protein ligands that are specific to DNA motifs are also being developed.³⁷ In work involving the in vivo cloning of artificial DNA nanostructures, the existence of immobile Holliday junctions in *Escherichia coli* has been implied.³⁸ As biological materials, DNA motifs also provide useful substrates to study molecular processes such as supercoiling, crossover isomerization and structure-specific protein binding.^{39,40} Thus, understanding the structural and thermodynamic characteristics of DNA motifs could lead to ascertaining the biological implications of these structures."*

10) Optional: the authors could mention in the discussion of Fig. 1 that the switchback DNA does not stain as strongly as the ds complexes, paving the way for the later work with small molecule binding.

Thank you for this suggestion. We've now added this information in the text pertaining to Fig. 1.

"We observed that the switchback DNA structure was not stained on the gels as much as the conventional duplex (see later section on small molecule binding to switchback DNA and conventional duplex)."

Reviewer 2:

In this manuscript, "Unusual Structural Properties of Switchback DNA", the authors present a detailed characterization of switchback DNA compared to canonical B-DNA duplexes. This synthetic DNA motif, discovered many years ago, is of interest due to its potential use in nucleic acid nanotechnology applications. The authors used a variety of techniques (in silico and in vitro) to characterize properties of switchback (relative to B-DNA duplex substrates) such as thermal stability, biostability (as assessed by digestion by several nucleases), small molecule binding experiments, cell viability, and immune responses. Their results indicate that there are some interesting differences in these properties between switchback DNA and B-DNA duplexes.

This is a timely study that has the potential to have a positive impact in guiding the development of new applications for synthetic DNA motifs in nanotechnology. The authors have provided a variety of approaches to draw their conclusions. However, some of the conclusions drawn are not completely

convincing with the data as currently presented. Below are listed several major concerns to be addressed by the authors.

Thank you for your positive evaluation of our manuscript and noting that the study is timely and would have a positive impact on developing synthetic DNA motifs. We've addressed your concerns in the manuscript and provide detailed responses below.

1) While this is a detailed characterization of switchback DNA, the authors may want to better explain the novelty and potential impact of the work on the field. For example, the work in cells is minimal and the computational simulation and prediction studies to suggest switchback DNA forms a biologically relevant alternative structure involved in repeat-expansion diseases is not convincing without data providing some type of biological evidence (see comment #7 below).

Thank you for the suggestion. We've added a new paragraph in the introduction to better place our work in the context of DNA nanotechnology. Our discussion paragraph also portrays this importance and relevance to DNA nanotechnology (both paragraphs provided below). Detailed responses to comments on cell and biological work are provided in later points.

In introduction:

*"The motifs used in the field of DNA nanotechnology were initially inspired by structures that exist in nature.²⁹ For example, immobile 4-way junctions were inspired by Holliday junctions.³⁰ Double crossover molecules^{8,31} have been proposed as intermediates in recombination,³² and have been demonstrated to be involved in meiosis.³³ While naturally occurring motifs can guide the design of DNA nanostructures, studying the properties of basic DNA motifs could provide insights into biological phenomena as well. There are several examples for this. PX DNA has been suggested to be involved in interactions between homologous segments of supercoiled DNA in processes such as recombination and repair.^{34,35} Further, proteins that bind to PX DNA in a structure-dependent fashion have been isolated,³⁶ and protein ligands that are specific to DNA motifs are also being developed.³⁷ In work involving the in vivo cloning of artificial DNA nanostructures, the existence of immobile Holliday junctions in *Escherichia coli* has been implied.³⁸ As biological materials, DNA motifs also provide useful substrates to study molecular processes such as supercoiling, crossover isomerization and structure-specific protein binding.^{39,40} Thus, understanding the structural and thermodynamic characteristics of DNA motifs could lead to ascertaining the biological implications of these structures."*

In discussion:

"In DNA nanotechnology, switchback DNA structure can be combined with sticky ends to create double-crossover-like structures that are used in 3D assemblies.⁴² Larger constructs based on switchback DNA are possible but may require chemical functionalization (such as 3'-3' or 5'-5' linkages) to allow for strand connectivity since the two strands in the structure are parallel. The left-

handedness of switchback DNA can be an advantage in some cases. In topological studies of catenanes and knots, structures constructed using conventional right-handed B-DNA duplexes have negative nodes,⁸⁷ and require the use of left-handed Z-DNA or L-DNA to produce a positive node. Incorporation of Z-DNA requires specific sequences and ionic conditions while L-DNA requires expensive chemical synthesis. In such cases, use of a switchback structure will provide a positive node due to its left-handedness. Further, switchback complementarity could be used to connect DNA motifs as an alternate for sticky ended cohesion, such as with bubble-bubble cohesion⁸⁸ or paranemic cohesion.⁸⁹ The requirement of Mg^{2+} to form stable switchback DNA could also be useful in ion-dependent assembly of DNA nanostructures, similar to DNA nanostructures constructed through loop-loop interactions at high Mg^{2+} concentrations.⁷⁹ Overall, our study provides a deeper understanding of the switchback DNA structure with utility in nanotechnology to fold DNA into biostable and functional structures for different applications, and presents a biological perspective for the DNA motif with possible implications in repeat expansion disorders."

2) The data presented in Figure 1 are a bit confusing and/or not completely convincing. The authors state that the switchback DNA migrated in non-denaturing polyacrylamide gels in a position similar to that of the B-DNA duplex. So how is this evidence for structure formation? They then designed a longer duplex complement so that the two substrates would migrate differently due to their lengths but again...how does this distinguish between the 2 structures rather than just the 2 sequences? The authors need to clarify their rationale and interpretation of the data and/or provide a different approach to better demonstrate the formation of switchback DNA to strengthen their conclusions.

The complex formed by the two component strands of switchback DNA shows a single band that corresponds to the size of a 2-stranded complex. No bands corresponding to the single strands are seen. Further characterization of this complex using UV-melting studies shows that it is a unique complex that melts with a T_m distinct (and less than) from the typical duplex. Moreover, the crystal structure of the switchback DNA with this sequence has been reported by Mao et al (ref 42 in the manuscript), providing additional confirmation on switchback DNA formation for these particular sequences.

Based on your suggestion, we also added another approach to show the formation of switchback DNA (data added in Fig. 2c-d). We performed a fluorescence experiment, where we modified the 5' terminus of strands X with fluorescein (fluorophore) and 5' termini of strands Y and Z with Iowa Black (quencher), and showed that formation of switchback DNA (complex XY) leads to fluorescence quenching.

The use of the longer duplex complement was mainly to distinguish both structures on a gel, and we use this design in later experiments on the analysis of structural preference during and post assembly (Fig 3). To avoid confusion, we've retained this data only in the SI and revised the text to explain this rationale better.

New experiment and text:

“To further demonstrate the formation of switchback DNA, we performed fluorescence spectroscopy experiments. We modified the 5' end of strand X to contain a fluorescein dye, and the 5' ends of strands Y and Z to contain Iowa Black quencher (Fig. 2c). Formation of conventional duplex (complex XZ) places the fluorophore and quencher on opposite termini of the duplex, resulting in a higher fluorescence. In contrast, the fluorophore and quencher are in close proximity in the switchback DNA (complex XY) due to the parallel orientation of the strands, causing a 37% reduction in fluorescence (Fig. 2d).”

Fig. 2. (c) Fluorescence experiments showing the formation of switchback DNA with strands in parallel orientation. (d) Percent fluorescence reduction during switchback DNA formation.

3) Related to comment #2 above, in Fig. S6, the authors state the CD signature of switchback DNA was similar to that of the B-DNA duplex, “confirming that the half-turns in the switchback DNA resemble a typical B-DNA structure despite the overall left-handed nature of the switchback structure”. This is not solidly convincing because if the CD signatures are the same, then what evidence is there to support the formation of the switchback structure under the conditions of the assay? The authors should provide a better rationale for this experiment and/or provide additional evidence using an orthogonal assay, which would strengthen the conclusions drawn in this study.

We draw our conclusions based on the several different experiments we had performed as well as prior work on the structure. Our results are consistent with the prior report by Chengde Mao’s group which shows similar gel-based analysis to demonstrate the formation of switchback DNA (ref 41 in manuscript).

In switchback DNA, the sugar-phosphate backbone traces a left-handed helical path, however, the nucleobases are stacked in a right-handed helix within the half-turn domains. The CD signatures originate from these base stacks. The half-turns with 6 base pairs are stacked like a B-DNA, as seen in the reported crystal structure (ref 42). Further, our CD signatures are similar to those described by Mao’s group thus showing that the conclusions we draw about the formation of switchback DNA are

valid. We've added additional text to clarify this along with the appropriate citation (new text below). We would like to point out that the way these sequences are designed, they can only form a switchback structure and not a conventional duplex if all the bases are to be paired.

However, we agree with your point about additional validation and designed a fluorescence-based experiment to show that the strands are parallelly oriented in a switchback structure (quenched fluorescence) in contrast to the antiparallel orientation of strands in a conventional duplex (higher fluorescence). This additional experiment is included in the new Fig. 2c-d and provided below for convenience.

New experiment and text:

"To further demonstrate the formation of switchback DNA, we performed fluorescence spectroscopy experiments. We modified the 5' end of strand X to contain a fluorescein dye, and the 5' ends of strands Y and Z to contain Iowa Black quencher (Fig. 2c). Formation of conventional duplex (complex XZ) places the fluorophore and quencher on opposite termini of the duplex, resulting in a higher fluorescence. In contrast, the fluorophore and quencher are in close proximity in the switchback DNA (complex XY) due to the parallel orientation of the strands, causing a 37% reduction in fluorescence (Fig. 2d)."

Fig. 2. (c) Fluorescence experiments showing the formation of switchback DNA with strands in parallel orientation. (d) Percent fluorescence reduction during switchback DNA formation.

4) The mismatch tolerance experiments using UV melting (Fig. 1k) are more convincing than the those using PAGE (Fig. 1i); however, the results from the 2 different types of experiments seem to differ in regard to the stability of the duplex DNA containing 2 mismatches. Could the authors comment on this apparent discrepancy?

We agree with your comment that the UV melting results more accurately portray the stability of the duplex structures with and without mismatches. We used gel analysis for the fully matched and mismatched sequences to test formation and not as a quantitative method for stability in this case

(also see new Supplementary Fig. 10). The gel presents a qualitative analysis of whether the complex is formed when mismatches are introduced in the sequence (based on the bands corresponding to the duplex and single strands). The UV-melting study provides the melting temperature to compare the stability of the complexes with and without mismatches.

We added some additional text to better describe the results:

“For each of these cases, we also designed duplex complements to form conventional duplexes with one or two mismatches. Non-denaturing PAGE analysis showed that the formation of conventional duplex was not affected by the presence of one or two mismatches, as indicated by the appearance of the band similar to the control structure without any mismatches (Fig. 2k, lanes 2-5). For the switchback structure, the presence of one mismatch slightly affected the assembly. We observed a band corresponding to the switchback DNA with a single mismatch (lane 7) that migrated closer to the position of the structure without any mismatches (lane 6) and is different from the band corresponding to single strands, indicating that the structure is still predominantly intact (Supplementary Fig. 10).”

5) In Fig. 4, it is not clear how the experiments were performed, but it seems to be done using purified enzymes and purified DNA substrates in vitro. Were the buffer conditions physiologically relevant? And while it did appear that the different nucleases tested had varying activity levels on the switchback vs. duplex substrate, would these differences still hold in cells? Would this make a difference regarding the use of switchback DNA in nanotechnology-based applications? The authors should discuss the relevance of these in vitro results. Further, if the authors demonstrated the relative stability of the switchback substrates to duplexes in cells, this would strengthen the impact of this study.

Thank you for these useful suggestions. The demonstration of relative stabilities of the structures inside cells would be ideal but for the current study and focus, we believe that the in vitro demonstration shows the idea in the suite of nucleases tested. As you suggested, we have revised the text to discuss the choice of buffer conditions and their relevance to DNA nanotechnology applications.

“We designed our experiments to be performed at the physiological temperature of 37 °C and in buffer conditions typically optimized for best enzymatic activity (Supplementary Table 4).

...

These results hold promise in DNA nanotechnology, where enhanced biostability could be introduced through structural design that occludes enzyme binding at specific loci and strategic placement of crossovers.^{23,57”}

6) In the cell viability and immune response experiments in HeLa cells (Fig. 5), the DNA substrates were incubated for 24-48 hours. Were the DNA substrates still intact after that time in cells? What was the percentage of DNA substrate remaining? The authors should determine the half-life of the various DNA substrates in cells. This would be informative and would strengthen the results on cell viability and immune responses shown.

We agree that understanding and knowing the half-life of DNA nanostructures in cells would be informative. However, such an analysis requires more effort on the design, synthesis and analysis inside cells and would be a project of its own. We have, however, added additional experiments showing that switchback DNA is intact at 37 °C in buffer for at least 48 hours. This result is included in the new Supplementary Fig. 20.

“As cell experiments are performed at the physiologically relevant temperature of 37 °C, we first tested the stability of switchback DNA and showed that the structure is intact for at least 48 h at 37 °C in buffer (Supplementary Fig. 20).”

Supplementary Fig. 20. Stability of switchback DNA at 37 °C for up to 48 hours.

7) The experiments on the “biological significance” (Fig. 5) of sequences that can form switchback DNA are not very convincing as presented. First, the authors show a preference for duplex formation over switchback DNA in Fig. 2, so what is the likelihood that these sequences would form switchback DNA in cells? The results shown are done using molecular dynamics and simulation studies. While this suggests that these structures may form in cells (and therefore, may have biological significance), the results are not convincing without some cell-based data with a biological outcome. Second, to suggest that switchback DNA may be involved in repeat-mediated diseases is quite speculative and premature. In order to be convincing, the authors would have to test the switchback substrates in comparison to the number of other alternative DNA structures that have been strongly implicated in repeat-mediated diseases in studies to determine structure formation, mutagenic potential, etc., etc.

We based our hypothesis on the repeat sequences we screened, which were found to have the sequence requirement for switchback formation. In the previously submitted version, we had screened several sequences and found that some repeat sequences satisfy the sequence requirement for switchback formation, and could theoretically form switchback DNA. We had also provided MD simulation studies to support this hypothesis.

In the revised manuscript, we've addressed this comment through the following updates:

- 1) We provide experimental evidence that repeat sequences can form switchback DNA in certain conditions in vitro (new Fig. 7 and Supplementary Fig. 24-25).
- 2) We've included additional analysis of MD simulations that compare switchback DNA and conventional duplexes formed by the same pair of sequences (Fig. 7d).
- 3) We've added additional discussion of prior literature and biological scenarios that support our hypothesis.

Based on these results, we believe this is the right time to speculate that switchback DNA may be one of the many structures in tandem repeat sequences. We've revised the text to better phrase our hypothesis and indicate that while we can't conclude that switchback DNA plays any role in repeat expansion diseases, our work shows that these structures could exist in these sequences and merit further investigation in the context of the molecular biology and tandem repeat sequences. Providing biological evidence would require a massive effort and would be a project (and a publication) of its own.

The revised text describing the new experiments and hypothesis are provided below:

Additional MD analysis:

*"Additionally, we used the Molecular Mechanics/Poisson Boltzmann Surface Area (MMPBSA) method⁶⁶ to calculate thermodynamic properties of the two configurations within the repeat sequence, from the simulation trajectories. The predicted ΔH difference between the duplex and the switchback conformations is ~ 11 kcal/mol (**Fig. 7d**) and primarily arises from altered electrostatics and solvation between the two conformations (**Supplementary Table 5**). This indicates that both conformations are energetically feasible even if the duplex conformation is slightly more favorable."*

New experiment with repeat sequences:

*"Next, we sought to test whether CAG/CTG repeats could adopt switchback conformation. Since the formation of antiparallel conventional duplex is preferred over parallel switchback DNA for the two complementary strands, we imposed geometric constraints on the strands by fixing them on a DNA double crossover (DX) scaffold that places the two sequences next to each other in the parallel orientation (**Fig. 7e**). The DX motif contains two adjacent double helical domains connected by two*

crossover points, a motif well characterized in our group and several others.^{8,67-71} We extended one strand from each double helical domain to create single stranded “interacting regions”, where both extensions are 5’ to 3’ from the DX scaffold. The design also contains a TT linker between the DX scaffold and the interacting regions to allow some flexibility. We fixed interacting region 1 to contain (CAG)₄ and varied the sequences in interacting region 2 to contain (i) (CTG)₄ that can form a switchback with (CAG)₄ in interacting region 1, (ii) (CTG)₂T₆ that can form only one domain of the switchback DNA or (iii) a T₁₂ sequence that cannot recognize (CAG)₄ (**Fig. 7e** and **Supplementary Fig. 24**). To study the interactions, we modified the 3’ end of interacting region 1 with the quencher Iowa Black and the 3’ end of interacting region 2 with fluorescein (**Fig. 7f**, sequence details in **Supplementary Fig. 24**). We first assembled the DX scaffold with the different combinations of interacting regions and validated assembly using non-denaturing PAGE and confirmed that the structures are isolated and that there are no higher-order aggregations caused by the single strand extensions on the DX scaffolds (**Fig. 7g**). We then obtained the fluorescence spectra for each of the annealed structures and quantified the peak fluorescence signals (**Fig. 7h**). The control structure (CAG)₄:T₁₂ showed a high fluorescence signal indicating that the two regions do not interact with each other. The fluorescence was slightly reduced (~9%) for the (CAG)₄:(CTG)₂T₆ pair indicating some interaction between the CAG and CTG domains. The (CAG)₄:(CTG)₄ version showed a 27% reduction in the fluorescence signal suggesting that switchback DNA formation quenched the fluorescence by bringing the fluorescein and quencher to close proximity due to base pairing between the two parallel strands. To confirm that the fluorescence quenching is indeed due to sequence-specific base pairing between the parallel strands, we added a 10-fold excess of unlabeled (CAG)₄ single strand. The unlabeled (CAG)₄ forms a conventional duplex with the (CTG)₄ in interacting region 2, thus displacing the fluorescein-labeled (CAG)₄ and restoring the fluorescence (**Supplementary Fig. 25**). These results provide preliminary data that indicate switchback DNA may form in repeat sequences under certain conditions. Alternate secondary or tertiary structures observed in repeat sequences generally occur in long stretches, are influenced by flanking sequences and can be transient.^{72,73} The DX scaffold is intended to provide this structural context for the two strands to interact. Based on these results and an analysis of direct repeat sequences, we hypothesize that short tandem repeat sequences with repeat lengths of 2, 3, 4, 6, and 12-nt may adopt switchback DNA as well as conventional duplex (**Supplementary Fig. 22**) and an equilibrium between the structures could have implications for repeat expansion diseases.”

Fig. 7. Potential role of switchback DNA in repeat sequences. (a) A pair of repeat sequences that can form a conventional duplex or switchback DNA. (b) Simulated structures of repeat sequences forming conventional duplex or switchback DNA. (c) Root mean square fluctuations (RMSF) of residues in the conventional duplex and switchback. (d) Calculated thermodynamic parameters of the conventional duplex and switchback DNA formed by repeat sequences. (e) Design of a double crossover (DX) motif scaffold to place two interacting regions in parallel orientation. The 3' ends of the interacting regions contain a fluorophore and a quencher to monitor interaction. (f) Hybridization of interacting region 1 containing a (CAG)₄ sequence with interacting region 2 containing a (CTG)₄ sequence forms a switchback DNA, reducing fluorescence signal. Two controls were used in this study. The interacting region 2 in the first control contains two repeating units of CTG and six thymines (denoted (CTG)₂T₆) and the interacting region 2 in the second control contains twelve thymines (denoted T₁₂). (g) Non-denaturing PAGE analysis of DX scaffold containing different interacting regions. (h) Fluorescence analysis of the interaction between two regions on the DX scaffold. (CAG)₄:(CTG)₄ shows less fluorescence indicating formation of a switchback structure. Data represent mean and standard deviations of experiments performed with three replicates.

Additional text revisions for biological relevance (in biological relevance section):

"The modular base pairing scheme of switchback DNA is distinct from the conventional end-to-end base pairing model. While molecular recognition between nucleic acids mainly occurs via the conventional base pairing model, it is conceivable that the switchback DNA mode of interaction may

also exist in certain structural contexts between pairs of sequences that allow such pairing. We hypothesize that switchback DNA could be a potential alternate structure in short tandem repeat sequences. That is, a pair of sense and antisense strands containing tandem direct repeat sequences could form either the switchback DNA or a conventional duplex (**Fig. 7a**). We screened various direct repeat sequences involved in repeat expansion diseases and observed that in some cases, a given sequence could have the same switchback complement and duplex complement (**Supplementary Fig. 22**)."

...

"Our hypothesis that switchback DNA could be an alternate secondary structure in repeat sequences is based on the fact that long tracts of repeat sequences are known to adopt several alternate structures.⁷⁴ The formation of such alternate structures in repeat sequences is dependent on DNA concentration, repeat length, negative supercoiling stress and flanking sequences.^{72,73} Our experiment using a DX scaffold suggests that under specific conditions, even short (CAG)₄:(CTG)₄ repeat sequences could form switchback DNA (**Fig. 7h**). Since the energy required for the formation of most alternative DNA secondary structures decreases as the repeat length increases,^{75,76} the propensity to form switchback DNA may increase with increasing number of repeats. Thus, switchback DNA could occur in long tracts of repeat sequences in the conformationally dynamic genomic DNA although conventional duplex is the preferred form in free oligonucleotides with two half-turn domains. Collectively, this information suggests that switchback DNA could potentially have a biological role in repeat sequences, but further work is needed to validate this hypothesis. The transient and dynamic nature of alternate structures, along with their very low occurrence (0.01 to 1% of total DNA in some cases⁷⁷) also makes it a challenge to detect these structures.⁷⁸ It is of note that PX DNA, another synthetic motif, has been shown to form in homologous segments of superhelical DNA³⁴ and proteins that structure-specifically bind to PX DNA have been found.^{36,37}"

In discussion:

"In addition to the assembly and characterization of switchback DNA, we explored the implications of the unique modular base pair complementarity paradigm of switchback DNA in a biological context. It is known that under certain conditions, long tracts of repeat sequences tend to adopt alternate secondary structures.⁷⁴ The sequence requirement for formation of the switchback DNA structure is satisfied by short tandem repeat sequences that are present widely in the eukaryotic genome.

...

The secondary structure formation can also be highly dynamic and interchangeable. For example, the purine-rich strand from an H-DNA-forming sequence can fold back to form a triplex structure at neutral pH in the presence of divalent cations such as Mg²⁺. However, under acidic conditions, the cytosines can be protonated, and the pyrimidine strand can serve as the third strand in the triplex structure.^{85,86} A similar case could exist for two sequences that can hybridize to form either conventional duplex or switchback DNA under specific conditions."

Reviewers' Comments:

Reviewer #1:

Remarks to the Author:

The reviewers have satisfactorily addressed almost all the points I raised.

In the discussion of the relative stability of duplex and switchback DNA the authors now say 'The predicted ΔH difference between the duplex and the switchback conformations is ~ 11 kcal/mol (Fig. 7d) and primarily arises from altered electrostatics and solvation between the two conformations (Supplementary Table 5). This indicates that both conformations are energetically feasible even if the duplex conformation is slightly more favorable.'

This confused me because the difference between the two configurations sounds quite large, which meant that it seemed wrong to describe one as only 'slightly more favorable' than the other. However, I then spotted that the ΔH values were around 90-100kcal/mol, which does indeed show that both conformations are feasible. I would recommend re-framing this discussion for clarity. Maybe the authors could open by quoting the numbers that show both conformations are feasible before discussing the difference and noting that the duplex is more favourable?

Also: is it possible to calculate an estimated ratio of duplex/switchback based on the ΔH values?

Finally: it is not clear whether the supporting figures in the response to reviewers are intended for publication or not. Personally I think they are helpful and should be published alongside the paper, particularly Supporting Figure 2.

Dr Katherine Dunn, 7th May 2024

Reviewer #2:

Remarks to the Author:

While the potential biological roles of the DNA substrates in this study are still in question, and not well addressed here, the authors have adequately addressed most of my other concerns.

Response to reviewers

The unusual structural properties and potential biological relevance of switchback DNA

Dear Reviewers,

Thank you for taking the time and effort to review our manuscript. We appreciate your comments and constructive criticisms and are glad that we have addressed your concerns to the best of our abilities. We have made additional revisions to address your comments.

Reviewer 1:

The reviewers have satisfactorily addressed almost all the points I raised.

In the discussion of the relative stability of duplex and switchback DNA the authors now say 'The predicted ΔH difference between the duplex and the switchback conformations is ~ 11 kcal/mol (Fig. 7d) and primarily arises from altered electrostatics and solvation between the two conformations (Supplementary Table 5). This indicates that both conformations are energetically feasible even if the duplex conformation is slightly more favorable.'

This confused me because the difference between the two configurations sounds quite large, which meant that it seemed wrong to describe one as only 'slightly more favorable' than the other. However, I then spotted that the ΔH values were around 90-100kcal/mol, which does indeed show that both conformations are feasible. I would recommend re-framing this discussion for clarity. Maybe the authors could open by quoting the numbers that show both conformations are feasible before discussing the difference and noting that the duplex is more favourable?

Also: is it possible to calculate an estimated ratio of duplex/switchback based on the ΔH values?

Thank you for this suggestion. We have now rephrased the sentence for clarity. We had a similar thought on the ratio of duplex/switchback but unfortunately, we can't calculate that based on this analysis.

"The predicted ΔH values for the duplex and the switchback conformations differ only by $\sim 12\%$ (Fig. 7d) and primarily arise from altered electrostatics and solvation between the two conformations (Supplementary Table 5)."

Finally: it is not clear whether the supporting figures in the response to reviewers are intended for publication or not. Personally I think they are helpful and should be published alongside the paper, particularly Supporting Figure 2.

We provided the supporting figures to respond to reviewers' comments. We choose to leave them in the response so as to not distract the audience from the main context of the paper.

Reviewer 2:

While the potential biological roles of the DNA substrates in this study are still in question, and not well addressed here, the authors have adequately addressed most of my other concerns.

We are glad that the reviewer feels we adequately address the concerns. We have further limited the claims made on the biological roles of switchback DNA. We've made several changes throughout the manuscript and provide a few points here:

Deletions:

*"Alternate secondary or tertiary structures observed in repeat sequences generally occur in long stretches, are influenced by flanking sequences and can be transient.^{72,73} The DX scaffold is intended to provide this structural context for the two strands to interact. Based on these results and an analysis of direct repeat sequences, we hypothesize that short tandem repeat sequences with repeat lengths of 2, 3, 4, 6, and 12-nt may adopt switchback DNA as well as conventional duplex (**Supplementary Fig. 22**) and an equilibrium between the structures could have implications for repeat expansion diseases."*

"Thus, switchback DNA could occur in long tracts of repeat sequences in the conformationally dynamic genomic DNA although conventional duplex is the preferred form in free oligonucleotides with two half-turn domains."

Addition: (in the discussion, penultimate paragraph)

"We have shown that the primary sequence requirement for the formation of switchback DNA is satisfied by repeat sequences, proximal placement of the complementary strands in parallel orientation may favor its formation, and when folded into switchback form, in silico, the repeat sequences can remain stable. However, further investigations are necessary to demonstrate the occurrence of the switchback form in the repeat sequences in vivo. We believe that the possible existence of switchback DNA in the genome warrants exploration, for it allows us to understand better the grammar and syntax of the molecular language of nucleic acids in biology."

Edits:

(1) Previous version: "Since the energy required for the formation of most alternative DNA secondary structures decreases as the repeat length increases,^{75,76} the propensity to form switchback DNA may increase with increasing number of repeats."

Current version: "Since the energy required for the formation of most alternative DNA secondary structures decreases as the repeat length increases,^{75,76} the potential role of switchback DNA is better considered in longer stretches of repeat sequences rather than the short two-domain structures characterized here."

(2) Previous version: "A similar case could exist for two sequences that can hybridize to form either conventional duplex or switchback DNA under specific conditions."

Current version: "We hypothesize that a similar case could exist for two sequences that can hybridize to form either conventional duplex or switchback DNA under specific conditions."

(3) Previous version: "These results provide preliminary data that indicate switchback DNA may form in repeat sequences under certain conditions."

Current version: "These results provide preliminary data that indicate switchback DNA may form in CAG repeat sequences under certain conditions."

Reviewers' Comments:

Reviewer #1:

Remarks to the Author:

The authors have fully addressed my remaining questions and I feel that the manuscript is now ready for publication in Nature Communications.

Katherine Dunn